There are amendments to this paper

# Reconstructing evolutionary trajectories of mutation signature activities in cancer using TrackSig

Yulia Rubanova[1,2], Ruian Shi[3], Caitlin F. Harrigan [1,2,4], Roujia Li[4], Jeff Wintersinger[1,2,4], Nil Sahin[2,4,5], Amit G. Deshwar[6], PCAWG Evolution and Heterogeneity Working Group, Quaid D. Morris [2,3]* & PCAWG Consortium

The type and genomic context of cancer mutations depend on their causes. These causes have been characterized using signatures that represent mutation types that co-occur in the same tumours. However, it remains unclear how mutation processes change during cancer evolution due to the lack of reliable methods to reconstruct evolutionary trajectories of mutational signature activity. Here, as part of the ICGC/TCGA Pan-Cancer Analysis of Whole Genomes (PCAWG) Consortium, which aggregated whole-genome sequencing data from 2658 cancers across 38 tumour types, we present TrackSig, a new method that reconstructs these trajectories using optimal, joint segmentation and deconvolution of mutation type and allele frequencies from a single tumour sample. In simulations, we find TrackSig has a 3–5% activity reconstruction error, and 12% false detection rate. It outperforms an aggressive baseline in situations with branching evolution, CNA gain, and neutral mutations. Applied to data from 2658 tumours and 38 cancer types, TrackSig permits pan-cancer insight into evolutionary changes in mutational processes.

[1] Department of Computer Science, University of Toronto, Toronto, ON, Canada. [2] Vector Institute, Toronto, ON, Canada. [3] University of Toronto, Toronto, ON, Canada. [4] Donnelly Centre for Cellular and Biomolecular Research, University of Toronto, Toronto, ON, Canada. [5] Department of Molecular Genetics, University of Toronto, Toronto, ON M5S 1A8, Canada. [6] The Edward S. Rogers Sr. Department of Electrical and Computer Engineering, University of Toronto, Toronto, ON, Canada. PCAWG Evolution and Heterogeneity Working Group authors and their affiliations appears at the end of the paper. PCAWG Consortium members and their affiliations appear online. *email: morrisq@mskcc.org

Somatic mutations accumulate throughout our lifetime, arising from external sources or from processes intrinsic to the cell[1,2]. Some sources generate characteristic patterns of mutations. For example, smoking is associated with G–T mutations; UV radiation is associated with C to T mutations[3–5]. Some processes provide a constant source of mutations[6] while others are sporadic[7].

One can estimate the contribution of different mutation processes to the collection of somatic mutations present in a sample through mutational signature analysis. In this type of analysis, single nucleotide variants (SNVs) are classified into 96 types based on the type of substitution and tri-nucleotide context (e.g. ACG–ATG)[2]. Mutational signatures across the 96 types were derived by non-negative matrix factorization in the previous work by Alexandrov et al.[2] . Many of the signatures are associated with known mutational processes including smoking[7], non-homologous double strand break repair[2], and ionizing radiation[8]. The activities of some signatures are correlated with patient age[6] and suggest their use as a molecular clock[9]. Thus, signature analysis can identify the DNA damage repair pathways that are absent in cancer, can predict prognosis[10], or guide treatment choice[11].

Formally, a "mutational signature" is a probability distribution over a categorical variable representing a mutation type, where each element is a probability of generating a mutation from the corresponding type[12]. Each signature is assigned an "activity" (also called "exposure") which represents the proportion of mutations that the signature generates. Activities for pre-defined signatures can be computed from the total mutational spectrum of a sample by using constrained regression[13,14].

Mutational sources can change over time[15–19]. Mutations caused by carcinogen activity stop accumulating when the activity ends[7]. Mutations associated with defective DNA damage repair, such as BRCA1 loss[1,2] will begin to accumulate after that loss. Recent analyses of sequencing data from single bulk samples have reported modest changes in signature activities between clonal and subclonal populations[9,20] based on groups of mutations identified by clustering their variant allele frequencies (VAFs). However, the accuracy of these methods relies heavily on the sensitivity and precision of this clustering, which is typically low[21,22] except, in some cases, in multi-region sequencing studies[15–19,23].

Here we introduce TrackSig, a new method to reconstruct signature activities across time without VAF clustering. We use VAF to approximately order mutations based on their prevalence within the cancer cell population and then track changes in signature activity that are consistent with this ordering.

We use realistic simulations and bootstrap analysis to help assess the accuracy of signature activity reconstructions under a variety of different evolutionary scenarios. Using TrackSig and Pan-cancer Analysis of Whole Genomes (PCAWG) dataset of 2658 cancers, we have previously demonstrated[24] that signature activities change often during the lifetime of a cancer. Here we show that these changes can often be a more sensitive indicator of new subclonal lineages than VAF clustering.

The PCAWG Consortium aggregated whole-genome sequencing data from 2658 cancers across 38 tumour types generated by the ICGC and TCGA projects. These sequencing data were re-analysed with standardised, high-accuracy pipelines to align to the human genome (reference build hs37d5) and identify germ-line variants and somatically acquired mutations, as described by PCAWG Network[25].

## Results

In this paper we perform the realistic simulations to evaluate TrackSig's performance at reconstructing signature activities, detecting the number of mutation clusters, and correctly placing the changepoints under different scenarios including violations of TrackSig's assumptions.

TrackSig was applied to the 2552 whole-genome sequencing samples with more than 600 SNVs contained within the white and grey lists of the PCAWG group. Here we provide methodological details of TrackSig's use on real data (PCAWG). The analysis of signature trends and relation of changepoints found by TrackSig to subclonal boundaries is described elsewhere[24]. Figure 1 shows examples of TrackSig trajectories for two tumour samples (breast cancer and leukaemia).

**Choice of mutation signatures**. By default, following Alexandrov et al.[2], we classify mutations into 96 types based on their three-nucleotide context. Point mutations fall into six different mutation types (i.e. $C \rightarrow [AGT]$ and $T \rightarrow [ACG]$) excluding complementary pairs. There are 16 ($4 \times 4$) possible combinations of the 5′ and 3′ nucleotides. Thus, SNVs are separated into 96 ($K = 16 \times 6 = 96$) types. However, TrackSig can use different mutation type labelling schemes, so long as the signatures and the mutation types are provided as input.

Within the context of PCAWG, we use the set of 48 single-base signatures (SBS) developed by PCAWG-Signature group. The first 30 of those signatures are slightly modified versions of original signatures defined by Alexandrov et al.[2,12] and have the same numbering and interpretation. The original 30 signatures are described at COSMIC (http://cancer.sanger.ac.uk/cosmic/signatures). Signature analysis methods, including TrackSig, fit activities for only a subset of the signatures. These signatures are called the "active" signatures. The activities for the non-active signatures are clamped to zero. For example, SBS 7 has been detected almost exclusively in skin cancers and likely describes mutations caused by UV light[2]. As such, it is only assigned active status in skin cancers. In our analysis, we use the active signatures reported by PCAWG-Signature group. For analyses based on COSMIC signatures, one can use active signatures per cancer type as provided on COSMIC website. TrackSig can also be used to automatically select active signatures, as described in a later section.

**Simulations**. We tested sensitivity of TrackSig in multiple error scenarios using simulated data with known ground truth. First, because signatures overlap in the mutation types that they can produce, we first test reconstruction accuracy when SNVs are accurately assigned to the time points to assess errors due to inability to correctly assign signature activity. We describe these non-parametric simulations in the next section.

In "Results" section, we assess reconstructions when the mutation ordering is inferred based on mutation VAF. In this scenario, reconstruction errors can occur when (i) cancer cell fraction (CCF) estimates are inaccurate and, (ii) there are two SNV clusters which overlap in CCF space but have different signature activity profiles. In the latter case, SNVs from both clusters will be located in the same or adjacent time point bins and will have a mixture of signature activity profiles from two clusters. To test reconstruction errors in these two scenarios, we produce clonal evolution simulations where we sample the VAF data from a clonal evolution model with binomial sequencing noise. To simulate the VAF detection limit for mutations imposed by somatic mutation calling, we remove any mutation with fewer than three variant reads.

Finally, as part of clonal evolution simulations, we assess model misspecification error by introducing violations of the assumptions of infinite sites and the relationship between CCF and timing of mutation occurrence. Also, in some simulations, we introduce mutations under neutral selection (i.e. neutrally

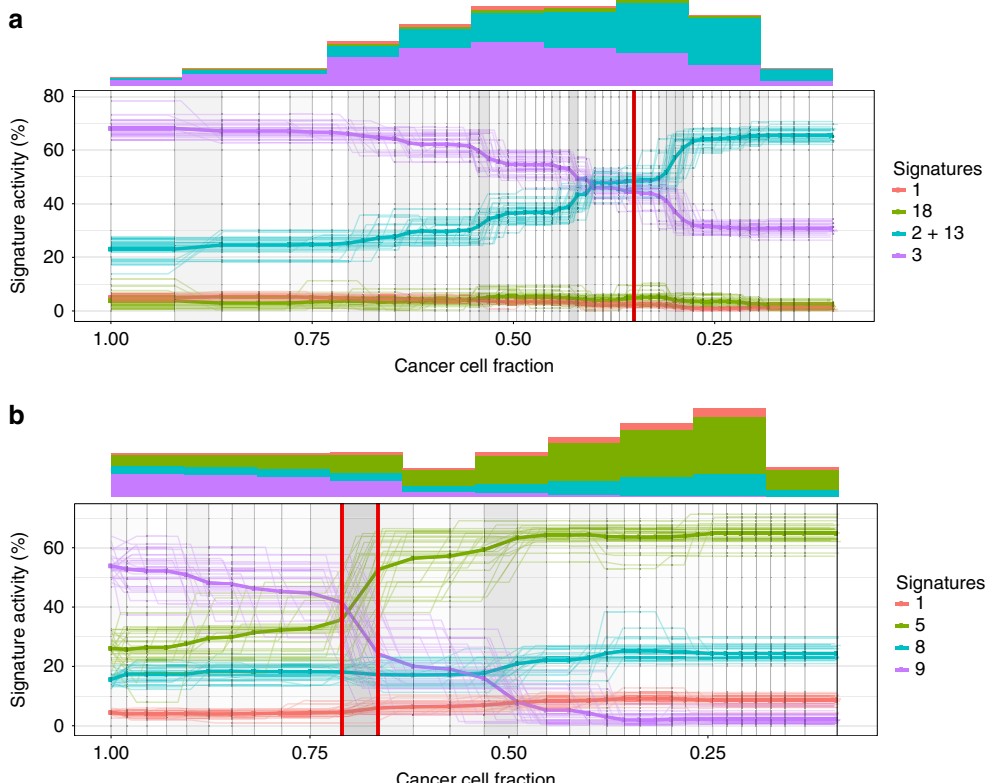

**Fig. 1 Signature activity trajectories for two samples.** Each plot is constructed from VAF data from a single tumour sample. Each line is an activity trajectory that depicts inferred activities for a single signature (y-axis) as a function of decreasing CCF (x-axis). The thin lines are trajectories from each of 30 bootstrap runs. The bold line depicts the mean activities across bootstraps. The vertical lines indicate time points in the original dataset, and are placed at the average CCF of their 100 associated mutations. Changes in activity trajectories are not necessarily aligned with vertical bars because mean CCFs of time points change across bootstraps. Frequency of changepoints between two vertical bars is indicated by shade, the darker shades indicate higher density of changepoints. Subclonal boundaries found by PCAWG consensus clustering[24] are shown in red vertical lines. These boundaries are not used in trajectory calculation and are only shown for comparison. Histograms show the mutation counts per signature in fixed width intervals of CCF. **a** Breast cancer sample. In clonal signatures remains constant with dominating signature 3 (associated with BRCA1 mutations). In the subclone activity to signature 3 decreases and is replaced by SNVs associated with APOBEC/AID (signatures 2 and 13). **b** Chronic lymphocytic leukaemia sample. Signature 9 (somatic hypermutation) dominates during clonal expansion and drops from 55% activity to almost zero in the subclone. Signature 5 compensates for this change.

evolving mutations). We computed the number and VAFs of these mutations using the model and effective mutation rates derived by Williams et al.[26,27] as detailed in Supplementary Note 3. As a baseline, we compare TrackSig's reconstruction error to the widely used strategy of first assigning SNVs to clusters based on VAF, then computing mutation signatures activities from the assigned mutations within each cluster.

**Non-parametric simulations**. In the non-parametric simulations, we test the ideal scenario when SNVs are correctly ordered and assigned to the time point bins. Here we want to access the ability of TrackSig to reconstruct signature activities from the distribution of mutation types and place changepoints at the correct locations.

Each simulation has 50 time points, each time point is a bin of 100 mutations. This corresponds to the average number of somatic mutations detected in PCAWG. Each sample also contains four active signatures. Two of those signatures are 1 and 5, which are nearly always active in the PCAWG samples. For the remaining two signatures, we test all 1035 possible combinations of the other 46 signatures.

We generate simulations with 0–3 changepoints that are placed randomly on the timeline. For each segment on the timeline, we sample signature activities from a uniform distribution over

activity vectors. Finally, we sample 100 mutation types per time point from the discrete distribution derived using the sampled activities as mixing coefficients for the four signatures.

Next, we run TrackSig on the simulated data and compare the reconstructed activity trajectories to the ground truth. We remove changepoints with small change, that is, where activities of all signatures change by <5% in reconstructed trajectories. This threshold is derived in "Results" section from permutation analysis.

We computed the absolute difference between predicted activities and the ground truth at each time point and take the median across all time points and all four signatures. We called this the median activity difference per simulation. On the simulations with no changepoints, the median of these median per simulation differences is 0.7%. On simulations with 1–3 changepoints, this median increases slightly to 2%. The cumulative distribution of the median per simulation differences is shown in Fig. 2.

For the PCAWG data, we report the maximum activity change (MAC) across activity trajectory[24] The maximum change is the difference between maximum and minimum activity across all time points in a sample. We also report the direction of change (down if maximum occurs before minimum and up otherwise). Here, we evaluate TrackSig's accuracy in these estimates on the simulated data. The MAC discrepancies between the estimated

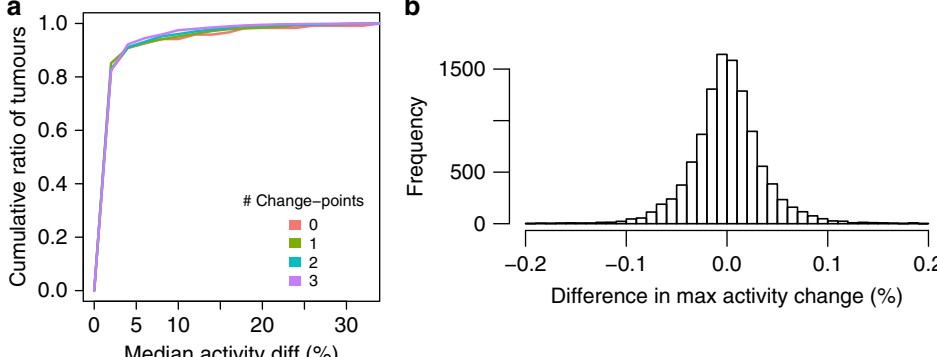

**Fig. 2 Results on non-parametric simulations. a** Median activity difference between the reconstructed trajectories and the ground truth. Lines correspond to the simulations with 0, 1, 2, or 3 changepoints. The median in computed across all signatures and time points in the sample. **b** Distribution of maximum activity change (MAC) discrepancies between between estimated activities and ground truth.

and ground-truth trajectories is <5% in 83.2% of cases across all signatures in all simulations (Fig. 2b).

To compare the direction of the activity change, we divide signatures into those with: decreasing activity, increasing activity, and no activity change (i.e. max absolute change is <5%). The direction of maximum change is consistent in 95.2% of all signatures across all simulations.

To compute the number of false positives and false negatives, we count a true positive detection if at least one of predicted changepoints occur within three time points of an actual one. A false negative is when no predicted changepoints are within three time points of an actual change. This criteria is identical to the one we use to evaluate whether a changepoint supports a subclonal boundary[24]. We deem a predicted changepoint a false positive if it occurs more than three time points away from the closest actual changepoint.

Tables 1 and 2 show the percentage of simulations where we observe a certain number of false negatives, and false positives respectively (see Supplementary Table 1 for additional results). On average, there are 0.12 false positives per simulation and 0.02 false negatives on average per simulation.

**Clonal evolution simulations**. Generating realistic simulated data requires making some assumptions about how tumours evolve. In this section, we simulate VAF data consistent with clonal evolution theory[28], with some small violations. Specifically, every mutation belongs to one of a small set of subclones. If the CCFs of each mutation could be estimated precisely, then mutation CCF data consistent with clonal evolution theory would have signature activities that are piece-wise constant functions of CCF. These VAF data are thus consistent with all previous work estimating signature activities.

We also include some simulations that violate the clonal evolution assumptions and test TrackSig's robustness to these violations. We performed six different simulations, described briefly here (see Supplementary Note 3 for details). We generated 100 simulations of each type.

First, we aim to evaluate false negative and false positive rates of identifying subclones via TrackSig. We simulated VAF data from (a) one clonal population and no subclones (b) one clonal population and one subclone with a variety of CCF values sampled from a uniform distribution, assuming a linear clonal tree. We sample the variant allele counts for the mutations in accordance to the cluster CCFs from a binomial distribution. We create simulations with four signatures—age-related signatures 1 and 5 and two other randomly-chosen signatures, which

**Table 1 False negatives rates in non-parametric simulations.**

| No. of true changepoints | | 0 | 1 | 2 | 3 |
|---|---|---|---|---|---|
| Avg no. of FN per simulation | | 0.0 | 0.008 | 0.038 | 0.058 |
| No. of false negative changepoints (FN) | 0 | 1 | 0.992 | 0.962 | 0.947 |
| | 1 | 0 | 0.008 | 0.038 | 0.049 |
| | 2 | 0 | 0 | 0 | 0.003 |

Each cell shows the proportion of simulations that have certain number of false negatives (normalized within the column). See main text for definition of positive and negative time points. The first row of the table shows the average number of false negatives per simulation

**Table 2 False positives rates in non-parametric simulations.**

| No. of true changepoints | | 0 | 1 | 2 | 3 |
|---|---|---|---|---|---|
| Avg No. of FP per simulation | | 0.130 | 0.128 | 0.118 | 0.116 |
| No. of false positive changepoints (FP) | 0 | 0.909 | 0.896 | 0.9 | 0.889 |
| | 1 | 0.06 | 0.083 | 0.087 | 0.106 |
| | 2 | 0.024 | 0.019 | 0.011 | 0.005 |
| | 3 | 0.005 | 0.002 | 0.002 | 0 |
| | 4 | 0.002 | 0 | 0.001 | 0 |

Each cell shows the proportion of simulations that have certain number of false positives (normalized within the column). See main text for definition of positive and negative time points. The first row of the table shows the average number of false positives per simulation

we will refer to as A1 and A2. The activities of A1 and A2 are sampled uniformly in each clone, under the constraint that at least one of them has a signature change of at least 30%. We sample mutation types from the signature mixture treating it as a multinomial distribution. We simulated mean read depths of 10, 30, and 100.

The performance of TrackSig was also assessed under conditions of neutral evolution. We sampled mutation VAFs as per the previous paragraph but we also added some neutrally evolving mutations to the clonal cluster. We determined the number of neutral mutations to add and sampled their VAFs according to the model from the Williams et al.[26] (see Supplementary Note 3). The signature activities for the neutral

mutations were the same as those of the other mutations associated with the clonal cluster.

To assess TrackSig's accuracy when the timeline does not reflect the ordering of acquistion of SNVs, we generated VAF data from a branched phylogeny. In branching simulation we generated VAF data assuming a branching clonal tree with two subclones. We force the sum of subclonal CCFs to be <1, otherwise the infinite sites assumption will be violated[1]. A later occurring subclone with a different signature activity profile has a higher CCF than a subclone with a profile matching the clonal fraction. We sample mutation VAFs and mutation types for the clusters similarly to the one- and two-cluster cases.

We next assess TrackSig's accuracy for reconstructing activity changes when SNV VAFs are affected by a copy number aberration (CNA). We generated VAF data with a clonal CNA gain affecting 10% of the SNV VAFs. In 5% of the mutations the CNA gain is affecting the mutant allele and in 5% the CNA gain is affecting the reference allele. This simulation is created similarly to the branching with three clusters. The difference is that we modify the probability of sampling a mutant allele to consider the altered mutant and reference copy numbers.

Finally, we create simulations with violation of infinite site assumption, where the same mutations independently occurred in two branched subclones. To model this, we set the CCFs of 3% of mutations to be equal to the sum of CCFs from the two subclones.

We compared results of TrackSig to the widely used approach of first clustering mutations by CCF and then inferring signature activities within each cluster[15–19]. We perform the clustering using SciClone[29]. We do a hard assignment of mutations to clusters detected by SciClone, and use DeconstructSigs[13] to estimate signature activities within these clusters. We report the results with two clustering methods in SciClone: Beta mixture model (BMM, default) and Beta-binomial mixture model (Binomial BMM). Note that the beta-binomial is an exact match to the noise model used in our data simulation, so we expect excellent performance from SciClone. We use the Beta model to simulate inaccuracies due to incorrect noise model specification.

**Simulation results**. We compared TrackSig and Sciclone + DeconstructSigs pipeline (hereafter SciClone for brevity, see Supplementary Note 4) against the ground truth in the simulation across the seven simulation types and depths 10, 30, and 100. First, we computed the median activity error over all mutations in the five types of non-neutral simulations, see Supplementary Fig. 7. In Supplementary Fig. 8 we compared the errors in the neutral one- and two-cluster cases. Across all depths, the majority (83.6%) of TrackSig reconstructions have <0.05 median error with ground truth versus 55.0% in SciClone. On average Track-Sig's activity error is 4.5%; SciClone's is 6.9% across all seven simulations. Measuring activity error using KL divergence gives similar results (Supplementary Fig. 4), and TrackSig's accuracy is relatively insensitive to bin size at multiple depths (see Supplementary Figs. 5 and 6)

We then compared methods based on their ability to detect subclones. Figure 3b shows the percentage of simulations when each method predicted correct number of subclones for depth 30 (see Supplementary Fig. 2 for depths 10 and 100). For this comparison, we used two different noise models with SciClone: the binomial-beta model which is an exact match to how the simulated data are generated, and the beta model, which is not and which was the model used for computing the reconstruction error above.

As expected, the SciClone binomial-beta performs nearly perfectly on the depth 30 simulations which match the

assumptions of this model (Fig 3b, one-cluster, two-clusters). However, the binomial-beta model is fragile, and performs poorly when its assumptions are violated (Fig. 3b, branching, cna, inf sites viol, one-cluster neutral evolution). The beta model does not perform well under the ideal situation, but it is better than the binomial model when the clonal evolution assumptions are violated. In contrast, TrackSig retains the ~10% false positive rate in calling subclones observed in the non-parametric simulations. TrackSig's high performance is maintained also in the neutral mutation simulations. In the other violation scenarios TrackSig has much better performance than either of the two SciClone noise models.

Supplementary Fig. 2 shows that at depth 100, TrackSig has ~90% accuracy in all scenarios except the two-cluster neutral evolution simulation. This scenario is particularly difficult because the clonal cluster is split with about 500 neutral mutations from the clonal lineage clustered at the VAF detection limit; so the mutation type distributions actually have two clear changepoints: one going from the clonal lineage to the subclonal one, and then another returning to the clonal (Fig. 4a). It may be possible to detect this error in post-processing (see "Discussion" Section). Note, however, that this simulation may not be representative of real data because, unlike other simulations[27], we are not simulating neutral mutations from the subclone. The depth 10 simulations are particularly challenging as well, because the VAF distributions of the clonal and subclonal clusters overlap substantially, making it difficult to detect multiple clusters. Here we see that TrackSig is an even more sensitive detector of a second cluster than the correct SciClone binomial model, see Fig. 4b. None of the methods do well in the branching, CNA, and infinite site violation scenarios because they require the detection of three clusters. Performance in the neutral evolution scenarios here matches that in the non-neutral ones because there are very few neutral mutations above the VAF detection limit. In summary, TrackSig is more robust to violations of the clonal evolution assumptions that are made by most subclonal reconstruction algorithms (see[23,27]).

**Methodology on real data**. We analyze the variation of signature activities on PCAWG data across time and across samples. We compute the maximum change of the signatures in each sample, which is simply the difference between maximum and minimum activity of the signature. To assess whether a signature change is statistically significant, we permute the mutations in each sample and run the trajectory estimation on the permuted set. Since permuted mutations are not sorted in time, we expect no change in the activity trajectories over time. The MAC that we observe on permuted set of mutations does not exceed 5% in any sample. Therefore, we only consider signature changes above 5% to be significant (Fig. 5).

**Bootstrapping**. We assess the variability in activity trajectories by performing bootstrap on the PCAWG data. We sample mutations with replacement from the original set and re-calculate their activities and changepoints. We perform 30 bootstrap runs for each sample. Figure 1 shows examples of bootstrapped trajectories from two samples (breast cancer and leukaemia).

Signature trajectories calculated on bootstrap data are stable. The mean standard deviation of activity values calculated at each time point is 2.9%. We also evaluate the consistency of signature changes across the entire activity trajectory: size of signature change and location of the changepoint. The mean standard deviation of the change in signature activity is 5.3% across the bootstraps. This standard deviation does not exceed 5% in 55.8%

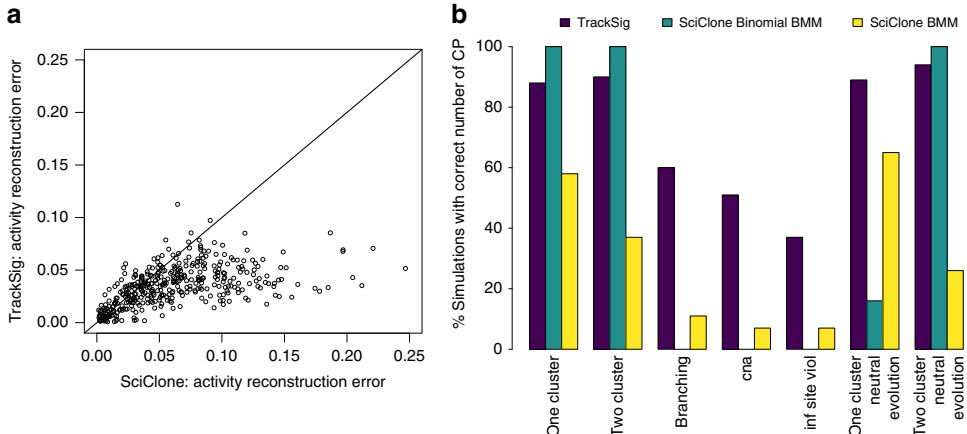

**Fig. 3 TrackSig and SciClone performance on clonal evolution simulations. a** Scatterplot of median activity errors (i.e. absolute activity difference) on all depth 30 simulations (see Supplementary Fig. 3 for depths 10 and 100). Mean activity error: TrackSig 3.5%, SciClone 6.2%. **b** Grouped barplot shows proportion of simulations where each method predicts the correct number of subclones for different simulation types as indicated on x-axis label. Different SciClone bars indicate different noise model selections. Results are for the simulations of average depth 30. Results for depths 10 and 100 are shown in Supplementary Fig. 2.

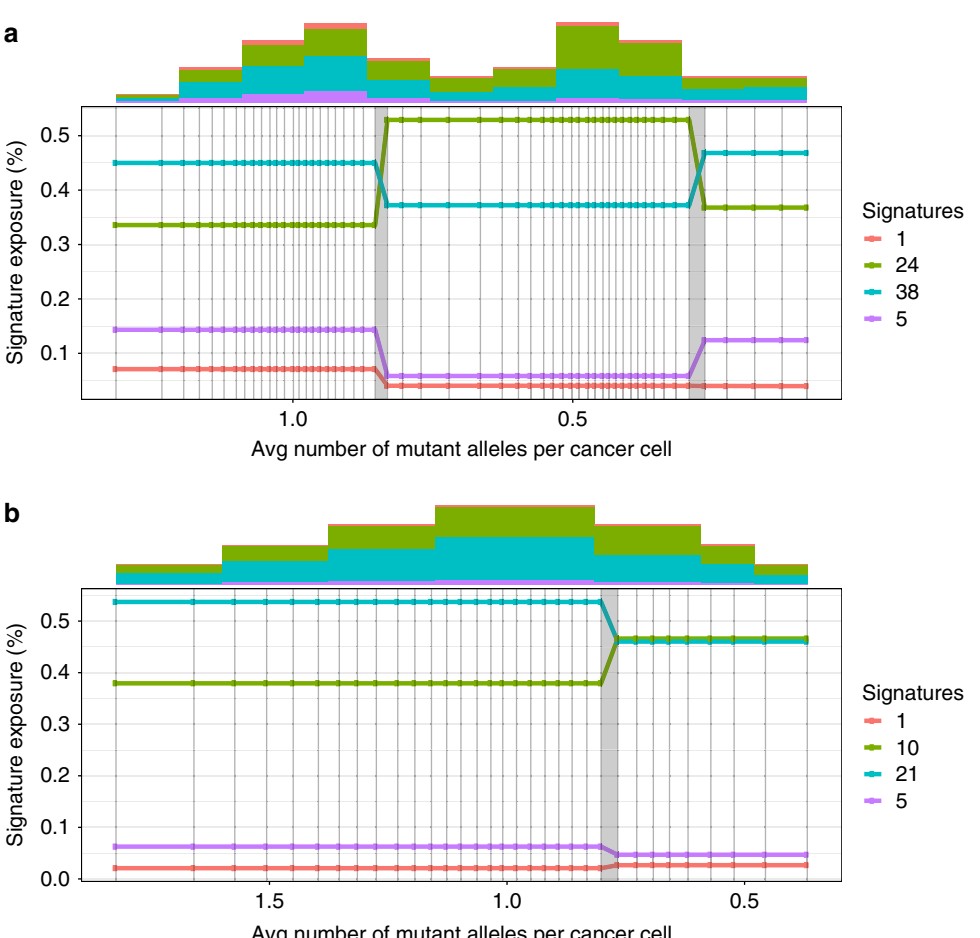

**Fig. 4 TrackSig reconstruction examples. a** Simulated data was generated with two clusters and clonal neutral mutations at read depth 100. TrackSig incorrectly places a changepoint before a cluster of neutral mutations from the clonal lineage near the VAF detection limit. However, because the signature activities match those in the clonal cluster, this error could be detected and corrected in post-processing. **b** Simulated data was generated with two clusters at read depth 10. TrackSig correctly identifies one changepoint. Although the simulation contains two clusters, there is only a single mode of CCF, thus making CCF-cluster-based detection of subclones impossible. However, the histogram on top shows that there are differences in mutation type distributions between the left and right tails, permitting TrackSig to correctly identify a changepoint. Both figures use an expanded x-axis that shows the whole spread of estimated CCF, this is indicated with a change in the x-label descriptor.

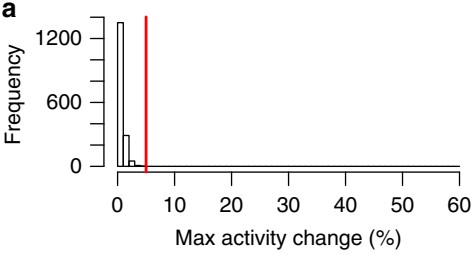
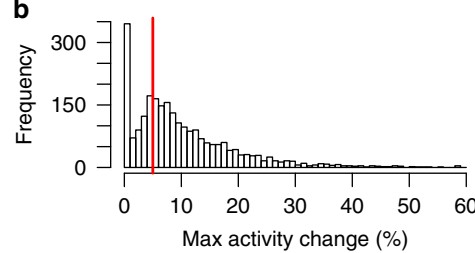

**Fig. 5 Maximum signature activity changes in PCAWG samples.** The red line shows the threshold of 5%, above which we consider changes to be significant. **a** Changes on random orders of mutations where we do not expect to see change in activities. **b** activity changes in TrackSig trajectories across all samples (on mutations sorted by CCF). Frequency axis shows the number of samples where we observe the certain activity change.

of samples (does not exceed 10% in 94.3% of samples, Supplementary Fig. 1).

In TrackSig the number of changepoints calculated during activity fitting does vary across bootstrap samples. We observe 1.02 standard deviation in the number of changepoints. To assess the variability in the location of the changepoints, we matched nearby changepoints between bootstrap samples and measured their average distance in CCF. Because the number of changepoints can change between samples, as a reference, we randomly choose one of the samples that has a number of changepoints equal to the median number of changepoints among all samples. Then, in all other bootstrap runs, we match each changepoint to the closest run in the reference. We found that location of the changepoints is consistent across bootstraps: on average, changepoints are located 0.093 CCF apart from the closest reference changepoint.

**Signatures with most changing activities.** As shown by Fig. 6, samples typically have only two or three signatures with high activities. These signatures are usually the most variable (up to 87.2% max change, 12% on average). Other signature have low-activity and remain constant. On average 3.6% of overall activity is made up of low-activity signatures (with activity <5%). Low-activity signatures most likely appear due to the uncertainty of our signature activity estimates. The mean standard deviation of signature activities is 2.9%, thus, we remove signatures with activity <5% as they are within two standard deviations of 0%.

**Trends in signature change per cancer type.** The majority of PCAWG samples have a signature change: 76.1% of samples have a max change >5% in at least one signature; 48.4% of samples have change >10%. However, the number of signature changes depends on the number of mutations in the sample. Out of samples with >10 time points only 26.3% of samples have a change >5% compared with 80.4% across the rest of the samples (see distribution on Fig. 7).

**Discussion**

TrackSig reconstructs the evolutionary trajectories of mutational signature activities by sorting point mutations according to their inferred CCF and then partitioning this sorted list into groups of mutations with constant signature activities. TrackSig estimates uncertainty in the location of the changepoints using bootstrap. TrackSig is designed to be applied to VAF data on SNVs from a single sample, however, it can be applied to either sorted lists of point mutations derived from subclonal reconstruction algorithms, or CCFs from a single cancer sample derived from methods which perform multi-sample reconstructions or subclonal CNA reconstructions.

Changepoints often correspond to boundaries between subclones[24]. In our simulations we show that TrackSig often better detects subclones than methods explicitly designed to find subclones, especially when there is a mismatch between the assumed and actual VAF generation process. By reconstructing changes in signature activities, TrackSig can potentially help identify DNA damage repair processes disrupted in the cell and, in doing so, help inform treatment[11].

Previous approaches estimate signature activities for a group of mutations without considering their timing (e.g. eMu[30] or deconstructSigs[13]). Therefore, the attempts to compare activity changes across evolutionary history have relied on pre-defined groups of mutations, such as those occurring before or after whole-genome duplications[7,9,31,32]; those classified as clonal or subclonal[1,9]; or those grouped in subclones via multi-region sequencing[15-19]. As such, the accuracy of these methods relies on (i) the accuracy in grouping mutations based on VAF—which is low with data from a single bulk sample[21]; and (ii) the existence of a small number of subclones or mutation groups within a sample, which is not true for neutrally evolving tumours[23,26,33].

In contrast, TrackSig uses the distributions of mutation types to group mutations, this permits more accurate reconstruction of signature activities than clustering mutations by VAF alone. Indeed, as our simulations demonstrate, not only are the signature activities more accurately reconstructed, but in some cases, TrackSig is a more sensitive detector of subclones. Furthermore, TrackSig makes fewer assumptions about the underlying VAF distribution, so it can be readily applied to data from neutrally evolving tumour populations[26,33]. Our simulations further demonstrate that TrackSig's reconstructions are less sensitive to model misspecification errors, such as violations of the infinite sites assumptions.

Clustering methods applied to VAFs from single bulk samples require high read depth for accuracy[21]. Indeed, due to this challenge, previous approaches have used multi-region sequencing[15-19,23,33-36]. In contrast, TrackSig can be deployed in a much larger range of settings. Separately, we report that TrackSig can detect subclones that are missed by VAF clustering methods[24]

Another important innovation of TrackSig is the use of CCF as a surrogate for evolutionary timing. Similar ideas have been used in human population genetics, where variant allele frequency to get relative order of mutations along the ancestral lineage[37]. In population genetics, allele frequency is calculated across individuals, while we calculate VAF across cell population within a single sample. In TrackSig we estimate CCF and reconstructions of clonal CNAs. In "Methods" section, we discuss the validity of using CCF as a surrogate for evolutionary time.

In TrackSig, the number of mutation types is provided as a parameter and is not fixed to 96 types. Because of this, it is straightforward to generalize TrackSig to reconstruct the activities

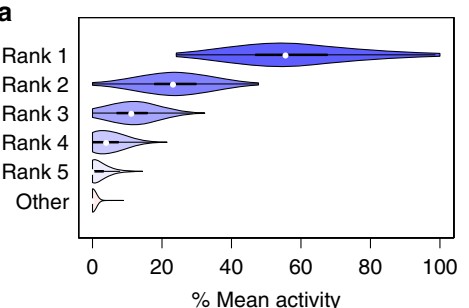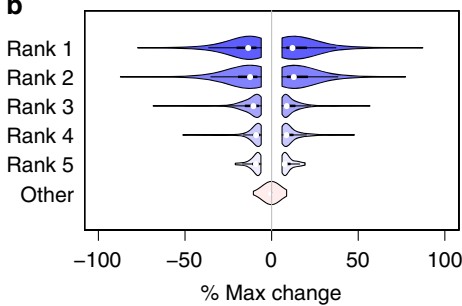

**Fig. 6 PCAWG signature changes by activity level. a** Mean signature activities ranked from the largest to the smallest within each sample in PCAWG data. Only the top five signatures with the highest activities in a sample are shown. **b** Maximum changes of signature activities for the corresponding signatures on plot (**a**). The changes below 5% are omitted.

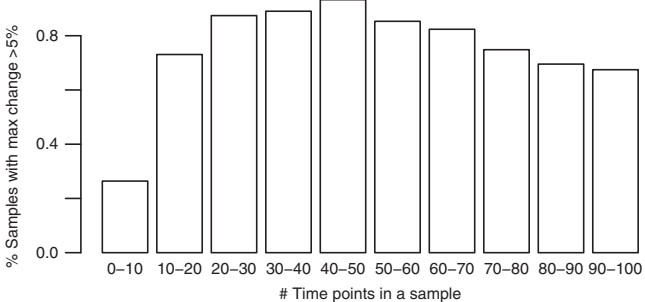

**Fig. 7 Frequency of activity change by number of mutations.** Proportion of tumours that have a significant change greater than 5% activity depending on the number of time points in a sample. Each bar corresponds to the range of number of time points in a sample; each time point contains 100 mutations.

of different mutation signatures or different mutations, so long as these mutations can be approximately ordered by their evolutionary time and each mutation can be classified into one of a fixed number of categories. In this paper, we ordered SNVs by decreasing CCF. This same strategy could be naturally extended to indels for which the infinite sites assumption is also valid. The infinite sites assumption should also be valid for structural variants (SVs) associated with well-defined breakpoints, thus permitting TrackSig to be used to track activities to recently defined SV signatures[32]. The CCFs of SVs can be estimated using the VAFs of split-reads mapping to their breakpoints[38]. Because they cover larger genomic regions, infinite sites is less valid for CNAs, although it is possible to approximately order clonal CNAs based on the inferred multiplicity of SNVs affected by them[9].

TrackSig also requires a pre-defined set of mutation signatures, each of which is a probability distribution over the mutation types. However, if these signatures are unavailable, they can be defined by non-negative matrix factorization, or Latent Dirichlet Allocation[39], if counts across mutation types are available from multiple cancer samples.

TrackSig can be applied to VAFs from bulk sequencing data from multi-region sequencing or longitudinal samples by simply running it on each sample separately. In preliminary experiments testing this approach we found broad consistency in the active signature selected, and in the signature activities of the clonal mutations in each sample. We observe with only 0.03% mean absolute activity difference (0.017 KL divergence) between signature activities of clonal cluster across different samples. See Supplementary Note 5 and Supplementary Fig. 9 for details.

For ease of presentation, we have assumed that ordering SNVs by CCF recovers the order in which they accumulated in the genomes of ancestral cells. However, this assumption is not critical for correct reconstruction of signature activity changes.

First, we have shown through bootstrap sampling and the clonal evolution simulations that errors in the estimation of SNV CCFs due to sampling noise have a limited impact on TrackSig's ability to estimate accurate activity trajectories. We have similarly shown that these activity trajectories are not impacted if a small fraction (3%) of the SNVs violate the infinite sites assumption.

However, these trajectories can be impacted by incorrect ordering of a large numbers of SNVs. These can occur in two ways. First, misordering can occur if a CNA changes the number of SNV allele's per cell. For example, daughter cells can fail to inherit SNVs in their mother cells due to a loss of heterozygosity (LOH). If a CNA reconstruction is available, TrackSig will correct for any detected clonal LOH when ordering SNVs, and will not attempt to order SNVs in regions affected by subclonal CNAs, thereby resolving this difficulty. However, if a CNA reconstruction is not available, or it is inaccurate, the accuracy of the activity trajectories can suffer. As such, we recommend only using TrackSig when CNA reconstructions are available and reliable.

Second, SNV ordering need not correspond to the time of acquisition when a single sample contains SNVs from subclones from different branches of the cancer phylogeny. In these circumstances, there is not a single linear order for the activities, and furthermore, late occurring subclones on a different branch can have higher CCF than earlier ones occurring in the sample. This situation also occurs when the sample contains a large number of neutrally evolving mutations from multiple subclonal lineages, as seen in the two cluster, depth 100 simulations. Note that such circumstances are rare in single biopsies[31] and that furthermore, a subclone can only be misordered if its CCF is <50% due to the Pigeonhole Principle[1], so the ordering by CCFs in guaranteed to be correct up until 50% CCF. However, even when these misorderings occurs, our simulations demonstrate that, with one exception, TrackSig's activity reconstructions, and estimation of number of subclones, are largely unaffected. Note, however, that the true complexity of tumour evolution with multiple subclones and high depth sequencing may confound the analysis in a way that is not assessed here, and so care must be taken interpreting results in tumours with complex clonal architectures.

Even in the rare circumstance that SNV misordering does occur, it may be possible to detect it, and interpret the activity changes correctly. For example, if late occurring but misordered SNVs manifest a more drastic change in signature activity, this misordering may be detectable by the presence of oscillations in the activity trajectories. To address this issue, when assessing overall change in signature activity, we computed the difference

between the lowest and highest activities for each signature. This difference will be consistent regardless of ordering.

The timelines reconstructed by TrackSig are computed with a fixed number of mutations in each bin. If overall rate of generating mutations in tumour was constant, our timeline would correspond to the real time. However, tumour mutation rate often accelerates throughout development[40,41]. Although the changing rate does not affect our analysis, the estimates of the pseudo-time might not be linearly related to real time.

Estimating changes in overall mutation rate is difficult. A possible way to correct for this is to adjust the timeline based on activities of signatures 1 and 5. Some report that signatures 1 and 5 operate as a cellular clock as the number of mutations contributed by these signatures is proportional to the age of the individual[6]. Determining the association between our pseudo-time estimates and real time is left for further investigation.

Our method TrackSig provides further insight how signature profile changes throughout tumour development. We show that through signatures analysis we can detect major events in tumour evolution, notably, transitions to a new subclone. Mutational signatures provide a unique way to recover tumour evolution path, track activities of mutational processes, adjust the treatment strategy and detect changes in therapy response.

## Methods

TrackSig is designed to be applied to VAF frequency data from a single, heterogeneous tumour sample. The method consists of two stages. First, we sort SNVs by their estimated CCF that we estimate using their VAFs and a CNA reconstruction of the samples. Next, we infer a trajectory of the mutational signature activities over the estimated ordering of the SNVs. We estimated activity trajectory for each signature as a piece-wise constant function of the SNV ordering with a small number of changepoints. These stages are described in detail below. Note that TrackSig does not rely on any methods for clustering mutations, such as phylogeny reconstruction.

**Ordering the SNVs**. No single evolutionary model can yet explain all of the observed VAF distributions in bulk tumour samples[21,23,26,27,42]. Using a CCF-based ordering of SNVs allows TrackSig to track changes in signature activity under a variety of such models. The clonal evolution model of cancer[28] posits that SNVs belong to one of a handful of subclones whose associated mutations all have the same CCF. Under this model, mutations from different subclones would occupy distinct regions of the CCF space, so if signature activities differ between subclones, the segments detected by TrackSig in CCF space would mark the presence of distinct subclones. Current neutral evolution models[21,26] assume SNVs to be unique and persistent (i.e. the infinite sites assumption) and SNVs with higher CCFs generally occur earlier in the tumour's evolution. Therefore, when SNVs are sorted in order of decreasing CCFs, TrackSig's trajectories track changes in signature activity over time.

In the following sections, we describe how the SNV VAFs are used to create a timeline to which TrackSig is applied. For ease of presentation, we will assume that the time of SNV occurrence increases approximately monotonically with position on the timeline. This interpretation is valid under the infinite sites assumption and either a neutral evolution model or a clonal one when all subclones are from the same branch, as is often the case in single samples[31]. TrackSig's reconstruction accuracy is tested on simulated data that is consistent with different evolutionary models and violations to these assumptions; and in the "Discussion" section, we discuss how to interpret TrackSig's reconstructions when the timeline is not a faithful representation of time of SNV acquisition.

**Estimating CCF**. Estimating a SNV's CCF requires both an estimate of its VAF and an estimate of the average number of mutant and reference alleles per cell at the locus where the SNV occurs. In TrackSig, we derive this estimate from a CNA reconstruction provided with the VAF inputs.

To account for uncertainty in a SNV's VAF due to the finite sampling, we model the posterior distribution over its VAF using a Beta distribution:

$$\text{VAF} \sim \text{Beta}(n_{\text{var}}, n_{\text{ref}}), \tag{1}$$

where $n_{\text{var}}$ is the number of reads carrying a variant, and $n_{\text{ref}}$ is the number of reference reads. To simplify the algorithm, and the subsequent sorting step, we sample an estimate of $\text{VAF}_i$ (VAF of SNV $i$) from this distribution and use that sample as a surrogate for the distribution in subsequent calculations. An advantage of this approach is that it gives us a single ordering. With a large number of SNVs, we expect little variability in the estimated activity trajectory due to uncertainty in

the VAFs of individual SNVs. With a smaller number of SNVs, multiple orderings can be sampled and the trajectories combined.

If no CNA reconstruction is available, TrackSig assumes that each SNV is in a region of normal copy number and TrackSig estimates CCFs in autosomal regions by setting:

$$\text{CCF}_i = \frac{2 \times \text{VAF}_i}{\text{Purity}}, \tag{2}$$

where Purity is the purity (i.e. proportion of cancerous cells) of the sample. If purity is not provided, TrackSig assumes Purity = 1.

If a CNA reconstruction is available, TrackSig uses it when converting from VAF to CCF. TrackSig assumes there is a maximum of one copy of the variant allele per cell, and thus estimates CCF by setting:

$$\text{CCF}_i = \frac{(2 + \text{Purity} \times (\text{CN}_i - 2))}{\text{Purity}} \times \text{VAF}_i, \tag{3}$$

where Purity is the purity of the sample, and $\text{CN}_i$ is the clonal copy number of the locus. If the clonal CNA increases the number of variant alleles per cell, this will lead to CCFs larger than one. As such, these cases are easily detected and corrected. Specifically, if the observed VAF is >50% due to finite sampling noise, or there is more than one variant allele per cell, the $\text{CCF}_i$ calculated above could be >100% which cannot be correct. As such, in most cases, we will set $\text{CCF}_i$ to be the minimum of Eq. (3) and 100%. When we do not do so, we will sometimes refer to these estimates as (estimated) "average number of mutant alleles per cell" to avoid confusion.

In regions of subclonal CNAs, estimating CCF requires a phylogenetic reconstruction in order to determine whether the subclonal CNA influences the number of variant alleles in the affected cells[22,43]. As such, when computing SNV ordering, by default, TrackSig filters SNVs in these regions out in order to avoid this time consuming operation. However, TrackSig can include these SNVs if provided with CCF estimates for them from methods that do consider subclonal CNAs[22,43,44].

TrackSig sorts SNVs in order of decreasing estimated CCF and uses the rank of the SNV in this list as a "pseudo-time" estimate of its time of appearance. Note that this estimate will have a non-linear relationship to real time, if the overall mutation rate can vary during the tumour's development. If some of the SNVs can be interpreted as clock mutations, an SNV's rank can be converted into an estimate of real time[9].

**Constructing a timeline**. To derive an estimate of the activity trajectory, TrackSig converts the SNV ordering into a set of time points with non-overlapping subsets of the SNVs. We do this for two reasons. First, stable estimation of signature activities requires a minimum number of mutations. By binning mutations into time points and requiring a minimum number of time points per segment, TrackSig enforces a minimum of 100 mutations per segment. Also, the time complexity of TrackSig scales with the number of time points. So by binning mutations, we can speed up TrackSig. By default, we set the bin size to 100 but the user can change this setting to as low as 1. As we show in "Results" section, TrackSig's signature activity reconstructions are relatively insensitive to the choice of bin size.

TrackSig first partitions the ordered mutations into bins and interprets each bin as one time point. The "timeline" of the cancer is the collection of the time points. TrackSig reports signature activity trajectories as a function of points in the timeline. We emphasize that TrackSig does not use any information about subclones when partitioning the SNVs and that TrackSig only uses CCFs for the SNV from a single sample.

**Computing activities of mutational signatures**. To estimate activity trajectories, TrackSig partitions the timelines into segments containing one or more time points. Within each of these segments, it estimates signature activities using mixture of discrete distributions. Full details of the model are provided in the Supplementary Note 1. In brief, TrackSig models each signature as a discrete distribution over the K types and it treats the mutation count vector over the K types as a set of independently and identically distributed samples from a mixture of the discrete distributions corresponding to each signature. By default, TrackSig uses single-base tri-nucleotide signatures[2] and $K = 96$, however TrackSig can use any mutation type labelling scheme, so long as it is given appropriate signatures as input. The mixing coefficients of these distributions are interpreted as their activities for the mixture model that produced the set of mutations. TrackSig fits these activities using the expectation-maximization algorithm (EM)[45], as done by other signature activity estimation methods[30].

**Detecting changepoints**. TrackSig identifies changepoints in the timeline where there are discernible differences in the activity of mutations in the time points before and after the changepoints. Specifically, the changepoints partition the timeline into segments of mutations with approximately constant activities. TrackSig fits activities for this set using EM algorithm as described above. This procedure generates piece-wise constant activity trajectories for each signature. To select changepoints, we adapt pruned exact linear time (PELT)[46], an optimal segmentation algorithm based on dynamic programming. We impose a complexity

penalty at each time point that is equivalent to optimizing the Bayesian Information Criteria (BIC) (see Supplementary Note 2 for details). To reduce variance in our estimates of the signature activities, we do not allow partitions to be <100 mutations.

We compute the BIC criteria the following way. Changepoints split the timeline into (# changepoints + 1) segments. In each segment, TrackSig fits the signature activities, which have to sum to one. Therefore there are (# signatures − 1) free parameters per segment, or (# changepoints + 1) · (# signatures − 1) free parameters in total. As such, BIC objective takes the following form:

$$BIC = -2ln\hat{L} + (\# \text{ changepoints} + 1) \cdot (\# \text{ signatures} - 1) \cdot ln(\# \text{ timepoints}), \quad (4)$$

where $\hat{L}$ is the likelihood of the current model.

**Correcting the timeline and segment count**. If the number of variant alleles per cell is increased by a clonal copy number change, TrackSig's CCF estimates might be >1. To correct for this, when displaying activity trajectories, it merges all the time points that have average CCF ≥ 1 into one time point. As such, the first time point can contain more than 100 mutations. To determine a signature activity at this new time point, TrackSig simply takes an average activity of all merged time points (those having CCF ≥ 1).

To compute the number of distinct subclones, we adjust the number of detected changepoints to correct for overlap in the CCF space of mutations from different subclones. Consider the case of two subclones whose mutations overlap substantially in CCF space. In this case, TrackSig might find three segments instead of two: one with signatures activities reflecting the first subclone; another with activities reflecting a mixture of the two subclones; and last with activities reflecting the second subclone. If this happens, then the direction of change of all signatures will be the same in the two changepoints. As such, when counting the number of distinct subclones, we treat each such pair of changepoints as one subclone boundary. Such a situation only occurs in 2.6% of 2552 PCAWG tumour samples to which we applied TrackSig; in 77% of those cases we remove a single changepoint.

**Bootstrapping to estimating activity uncertainty**. TrackSig estimates uncertainty in the activity estimates by bootstrapping the mutations and refitting the activity trajectories. Specifically, it takes the random subset of N mutations by sampling uniformly with replacement from the N unfiltered SNVs in the sample under consideration. Using the pre-assigned CCF estimates, we sort the SNVs in decreasing order, as above, re-partition them into time points and recompute activity estimates. The trajectories obtained from bootstrapped mutation sets have the same number of time points, however the average CCF for each time point can change. We use these bootstrapped trajectories to compute uncertainty estimates for the sizes of activity changes.

**Choosing active signatures**. Only a subset of signatures are active in a particular sample, and this subset is largely determined by a cancer type. For the analyses reported above, we use a set of active signatures provided by PCAWG[47], which contains a list of active signatures per sample (on average, four per sample). Frequent active signatures for each cancer type are available from a variety of sources[2,47]. We highly recommend using either these cancer-specific active signatures or deriving sample-specific active signatures using one of the procedures described in this section. We strongly discourage using TrackSig with a full set of signatures on a single sample, as many of the signatures overlap considerably, which can cause signature activity estimation errors due to this collinearity.

Here we evaluate three different ways to select the active signatures, all supported by TrackSig. The first strategy, "all-sigs", simply computes activity trajectories for all signatures. The second, "cancer-type-specific-sigs", uses all signatures reported as active in the cancer type under consideration. The final strategy, "sample-specific-sigs", first fits signature activities to the full set of mutation counts using an initial set of signatures, and sets the active signatures to be those with activities greater than a threshold (by default, 5%) in the initial fit. Then TrackSig computes activity trajectories only for the active signatures. In the following, we evaluate "sample-specific-sigs" when the initial set is "all-sigs", however, we suspect this approach will also work well with "cancer-type-specific-sigs" as the initial set. We evaluate each strategy by comparing the active signatures selected by TrackSig with those reported by PCAWG-Signature group on the PCAWG tumour set[47].

For "all-sigs", we used a set of 48 signatures and we found on average, 44.7% of overall activity assigned by TrackSig is assigned to the active signatures selected by PCAWG-Signature group. Each incorrect signature gets 1.3% of activity on average. In other words, the incorrect activity is widely distributed among the signatures. Using "cancer-type-specific-sigs" improves the correspondence to 68.7% of the total activity on average. This strategy reduces the initial set of potentially active signatures from 48 down to 12 on average (ranging from 4 signatures in Lower Grade Glioma to 24 signatures in Liver Cancer). Here, we observe that signature 5 and 40 are the most prevalent among the incorrect signatures, having the average activity of 14% and 12.6%, respectively in the samples where they are supposed to be inactive. Finally, if we use the "sample-specific-sigs" strategy starting with "all-sigs" as the initial set, we exactly recover the active signatures reported by PCAWG-Signature group.

Fitting either per cancer or per sample signatures results in more activity mass to be on the correct signatures and speeds up the computations. Therefore, we recommend choosing one of these instead of using activities from the full set.

**Reporting summary**. Further information on research design is available in the Nature Research Reporting Summary linked to this article.

## Data availability

Somatic and germline variant calls, mutational signatures, subclonal reconstructions, transcript abundance, splice calls and other core data generated by the ICGC/TCGA Pan-cancer Analysis of Whole Genomes Consortium is described here[25] and available for download at https://dcc.icgc.org/releases/PCAWG. Additional information on accessing the data, including raw read files, can be found at https://docs.icgc.org/pcawg/data/. In accordance with the data access policies of the ICGC and TCGA projects, most molecular, clinical and specimen data are in an open tier which does not require access approval. To access potentially identification information, such as germline alleles and underlying sequencing data, researchers will need to apply to the TCGA Data Access Committee (DAC) via dbGaP (https://dbgap.ncbi.nlm.nih.gov/aa/wga.cgi?page=login) for access to the TCGA portion of the dataset, and to the ICGC Data Access Compliance Office (DACO; http://icgc.org/daco) for the ICGC portion. In addition, to access somatic SNVs derived from TCGA donors, researchers will also need to obtain dbGaP authorisation.

## Code availability

TrackSig Code is available at https://github.com/morrislab/TrackSig. Code for generating simulation data is included in the Github repository. The core computational pipelines used by the PCAWG Consortium for alignment, quality control and variant calling are available to the public at https://dockstore.org/search?search=pcawg under the GNU General Public License v3.0, which allows for reuse and distribution.

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

# ARTICLE

17. Gerlinger, M. et al. Genomic architecture and evolution of clear cell renal cell carcinomas defined by multiregion sequencing. *Nat. Genet.* **46**, 225–233 (2014).
18. Harbst, K. et al. Multiregion whole-exome sequencing uncovers the genetic evolution and mutational heterogeneity of early-stage metastatic melanoma. *Cancer Res.* **76**, 4765–4774 (2016).
19. Yates, L. R. et al. Subclonal diversification of primary breast cancer revealed by multiregion sequencing. *Nat. Med.* **21**, 751–759 (2015).
20. Yates, L. R. et al. Genomic evolution of breast cancer metastasis and relapse. *Cancer Cell* **32**, 169–184.e7 (2017).
21. Griffith, M. et al. Optimizing cancer genome sequencing and analysis. *Cell Syst.* **1**, 210–223 (2015).
22. Deshwar, A. G. et al. PhyloWGS: reconstructing subclonal composition and evolution from whole-genome sequencing of tumors. *Genome Biol.* **16**, 35 (2015).
23. Sun, R. et al. Between-region genetic divergence reflects the mode and tempo of tumor evolution. *Nat. Genet.* **49**, 1015–1024 (2017).
24. Dentro, S. C. et al. Portraits of genetic intra-tumour heterogeneity and subclonal selection across cancer types. Preprint at https://doi.org/10.1101/312041 (2018).
25. The ICGC/TCGA Pan-Cancer Analysis of Whole Genomes Consortium. Pan-cancer analysis of whole genomes. *Nature* https://doi.org/10.1038/s41586-020-1969-6 (2020).
26. Williams, M. J., Werner, B., Barnes, C. P., Graham, T. A. & Sottoriva, A. Identification of neutral tumor evolution across cancer types. *Nat. Genet.* **48**, 238–244 (2016).
27. Williams, M. J. et al. Quantification of subclonal selection in cancer from bulk sequencing data. *Nat. Genet.* **50**, 895–903 (2018).
28. Nowell, P. C. The clonal evolution of tumor cell populations. *Science* **194**, 23–28 (1976).
29. Miller, C. A. et al. Sciclone: inferring clonal architecture and tracking the spatial and temporal patterns of tumor evolution. *PLoS Comput. Biol.* **10**, 1–15 (2014).
30. Fischer, A., Illingworth, C. J., Campbell, P. J. & Mustonen, V. Emu: probabilistic inference of mutational processes and their localization in the cancer genome. *Genome Biol.* **14**, R39 (2013).
31. Jamal-Hanjani, M. et al. Tracking the evolution of non-small-cell lung cancer. *N. Engl. J. Med.* **376**, 2109–2121 (2017).
32. Nik-Zainal, S. et al. Landscape of somatic mutations in 560 breast cancer whole-genome sequences. *Nature* **534**, 47–54 (2016).
33. Sottoriva, A. et al. A big bang model of human colorectal tumor growth. *Nat. Genet.* **47**, 209–16 (2015).
34. Gerlinger, M. et al. Intratumor heterogeneity and branched evolution revealed by multiregion sequencing. *N. Engl. J. Med.* **366**, 883–892 (2012).
35. Ling, S. et al. Extremely high genetic diversity in a single tumor points to prevalence of non-darwinian cell evolution. *Proc. Natl Acad. Sci. USA* **112**, E6496–E6505 (2015).
36. Caravagna, G. et al. Detecting repeated cancer evolution from multi-region tumor sequencing data. *Nat. Methods* **15**, 707–714 (2018).
37. Harris, K. & Pritchard, J. K. Rapid evolution of the human mutation spectrum. *Elife* **6**, e24284 (2017).
38. Cmero, M. et al. Inferring structural variant cancer cell fraction. *Nat. Commun.* https://doi.org/10.1038/s41467-020-14351-8 (2020).
39. Blei, D. M., Ng, A. Y. & Jordan, M. I. Latent dirichlet allocation. *J. Mach. Learn. Res.* **3**, 993–1022 (2003).
40. Alberts, B. *Essential Cell Biology*, Vol. 5.5 (Garland Science, 2010).
41. Wodarz, D., Newell, A. C. & Komarova, N. L. Passenger mutations can accelerate tumour suppressor gene inactivation in cancer evolution. *J. R Soc. Interface* **15**, 20170967 (2018).
42. Tarabichi, M. et al. Neutral tumor evolution? *Nat. Genet.* **50**, 1630–1633 (2018).
43. Roth, A. et al. PyClone: statistical inference of clonal population structure in cancer. *Nat. Methods* **11**, 396–398 (2014).
44. Jiang, Y., Qiu, Y., Minn, A. J. & Zhang, N. R. Assessing intratumor heterogeneity and tracking longitudinal and spatial clonal evolutionary history by next-generation sequencing. *Proc. Natl Acad. Sci. USA* **113**, E5528–E5537 (2016).
45. Dempster, A. P., Laird, N. M. & Rubin, D. B. Maximum likelihood from incomplete data via the em algorithm. *J. Royal Stat. Soc., Series B* **39**, 1–38 (1977).
46. Killick, R., Fearnhead, P. & Eckley, I. A. Optimal detection of changepoints with a linear computational cost. *J. Am. Statist. Ass.* **107**, 1590–1598 (2012).
47. Alexandrov, L. B. et al. The repertoire of mutational signatures in human cancer. *Nature* https://doi.org/10.1038/s41586-020-1943-3 (2020).

## Acknowledgements

We thank Pan-cancer Analysis of Whole Genomes (PCAWG) network, and in particular the PCAWG Evolution and Heterogeneity working group, for providing data, analysis and valuable input on this project. We would in particular like to highlight Peter Van Loo, Clemency Jolly, Stefan Dentro, David Wedge, Paul Boutros, Lydia Liu, and Moritz Gerstung who provided valuable feedback during the development of the TrackSig methodology. We acknowledge the contributions of the many clinical networks across ICGC and TCGA who provided samples and data to the PCAWG Consortium, and the contributions of the Technical Working Group and the Germline Working Group of the PCAWG Consortium for collation, realignment and harmonised variant calling of the cancer genomes used in this study. We thank the patients and their families for their participation in the individual ICGC and TCGA projects. We would like to acknowledge SciNet as part of Compute Canada for providing computational resources. This research was partially supported by an Natural Science and Engineering Research Council operating grant; an Associate Investigator award from the Ontario Institute of Cancer Research; and a subgrant from the Canadian Centre for Computational Genomics genomics technology platform funded by Genome Canada, all to QDM. It also received funding from the University of Toronto's Medicine by Design initiative, which in part of the Canada First Research Excellence Fund (CFREF) and the Compute the Cure gift from the NVIDIA foundation. QDM is a Canada CIFAR AI chair at the Vector Institute.

## Author contributions

Q.D.M. designed the project and supervised the study. Y.R. designed and implemented the method and performed the experiments. Y.R. and Q.D.M. wrote the manuscript with assistance from C.H. and RS. Y.R. and C.H. made figures. R.S. implemented PELT algorithm. R.L. performed the non-parametric simulations. C.H. and Y.R. performed clonal evolution simulations. C.H. implemented the SciClone+DeconstructSigs baseline. J.W. and A.D. provided assistance with tumour phylogeny reconstruction. N.L. wrote the script to extract tri-nucleotide counts. The PCAWG Evolution and Heterogeneity Working Group (co-led by Paul T Spellman, Peter Van Loo, and David C Wedge) provided critical feedback during that the development of TrackSig. The PCAWG Consortium, as whole, provided analysis of whole-genome sequencing data used herein, the mutational signatures, and feedback on the method. All authors read and approved the final manuscript.

## Competing interests

The authors declare no competing interests.

## Additional information

## PCAWG Evolution and Heterogeneity Working Group

Stefan C. Dentro[7,8,9], Ignaty Leshchiner[10], Moritz Gerstung[11], Clemency Jolly[7], Kerstin Haase[7], Maxime Tarabichi[7,8], Jeff Wintersinger[1,2,4], Amit G. Deshwar[6], Kaixian Yu[13], Santiago Gonzalez[11], Yulia Rubanova[1,2], Geoff Macintyre[14], David J. Adams[8], Pavana Anur[15], Rameen Beroukhim[10,16], Paul C. Boutros[12,17,18], David D. Bowtell[19], Peter J. Campbell[8], Shaolong Cao[13], Elizabeth L. Christie[19,20], Marek Cmero[20,21], Yupeng Cun[22], Kevin J. Dawson[8], Jonas Demeulemeester[7,23], Nilgun Donmez[24,25], Ruben M. Drews[14], Roland Eils[26,27], Yu Fan[13], Matthew Fittall[7], Dale W. Garsed[19,20], Gad Getz[10,28,29,30], Gavin Ha[10], Marcin Imielinski[31,32], Lara Jerman[11,33], Yuan Ji[34,35], Kortine Kleinheinz[26,27], Juhee Lee[36], Henry Lee-Six[8], Dimitri G. Livitz[10], Salem Malikic[24,25], Florian Markowetz[14], Inigo Martincorena[8], Thomas J. Mitchell[8,37], Ville Mustonen[38], Layla Oesper[39], Martin Peifer[22], Myron Peto[15], Benjamin J. Raphael[40], Daniel Rosebrock[10], S. Cenk Sahinalp[25,41], Adriana Salcedo[17], Matthias Schlesner[26], Steven Schumacher[10], Subhajit Sengupta[34], Ruian Shi[3], Seung Jun Shin[13,42], Oliver Spiro[10], Lincoln D. Stein[17], Ignacio Vázquez-García[8,37], Shankar Vembu[12], David A. Wheeler[43], Tsun-Po Yang[22], Xiaotong Yao[31,32], Ke Yuan[14,44], Hongtu Zhu[13], Wenyi Wang[13], Quaid D. Morris[2,3], Paul T. Spellman[15], David C. Wedge[9,45] & Peter Van Loo[7,23]

[7]The Francis Crick Institute, London NW1 1AT, UK. [8]Wellcome Trust Sanger Institute, Cambridge CB10 1SA, UK. [9]Big Data Institute, University of Oxford, Oxford OX3 7LF, UK. [10]Broad Institute of MIT and Harvard, Cambridge, MA 02142, USA. [11]European Molecular Biology Laboratory, European Bioinformatics Institute (EMBL-EBI), Cambridge CB10 1SD, UK. [12]University of Toronto, Toronto, ON M5S 3E1, Canada. [13]The University of Texas MD Anderson Cancer Center, Houston, TX 77030, USA. [14]Cancer Research UK Cambridge Institute, University of Cambridge, Cambridge CB2 0RE, UK. [15]Molecular and Medical Genetics, Oregon Health and Science University, Portland, OR 97231, USA. [16]Dana-Farber Cancer Institute, Boston, MA 02215, USA. [17]Ontario Institute for Cancer Research, Toronto, ON M5G 0A3, Canada. [18]University of California, Los Angeles, CA 90095, USA. [19]Peter MacCallum Cancer Centre, Melbourne, VIC 3000, Australia. [20]University of Melbourne, Melbourne, VIC 3010, Australia. [21]Walter and Eliza Hall Institute, Melbourne, VIC 3000, Australia. [22]University of Cologne, 50931 Cologne, Germany. [23]University of Leuven, B-3000 Leuven, Belgium. [24]Simon Fraser University, Burnaby, BC V5A 1S6, Canada. [25]Vancouver Prostate Centre, Vancouver, BC V6H 3Z6, Canada. [26]German Cancer Research Center (DKFZ), 69120 Heidelberg, Germany. [27]Heidelberg University, 69120 Heidelberg, Germany. [28]Massachusetts General Hospital Center for Cancer Research, Charlestown, MA 02129, USA. [29]Department of Pathology, Massachusetts General Hospital, Boston, MA 02114, USA. [30]Harvard Medical School, Boston, MA 02215, USA. [31]Weill Cornell Medicine, New York, NY 10065, USA. [32]New York Genome Center, New York, NY 10013, USA. [33]University of Ljubljana, 1000 Ljubljana, Slovenia. [34]NorthShore University HealthSystem, Evanston, IL 60201, USA. [35]The University of Chicago, Chicago, IL 60637, USA. [36]University of California Santa Cruz, Santa Cruz, CA 95064, USA. [37]University of Cambridge, Cambridge CB2 0QQ, UK. [38]University of Helsinki, 00014 Helsinki, Finland. [39]Carleton College, Northfield, MN 55057, USA. [40]Princeton University, Princeton, NJ 08540, USA. [41]Indiana University, Bloomington, IN 47405, USA. [42]Korea University, Seoul 02481, Republic of Korea. [43]Human Genome Sequencing Center, Baylor College of Medicine, Houston, TX 77030, USA. [44]University of Glasgow, Glasgow G12 8RZ, UK. [45]Oxford NIHR Biomedical Research Centre, Oxford OX4 2PG, UK

