## [Peer Review File · Nature Communications]

Reviewers' Comments:

Reviewer #1:

Remarks to the Author:

The authors introduce TrackSig, a method for inferring how mutation signatures have changed over time in individual tumors. The method takes as input (1) a set of SNVs with associated CCFs and (2) a set of active signatures, and outputs for each timepoint (a set of SNVs with similar CCFs) the abundance of each active signature. TrackSig is based on two key assumptions: (1) the tumor sample is composed of clones present from the same branch in the underlying phylogenetic tree, (2) CCF of an SNV is correlated with the time at which that SNV was introduced. The authors validate their method on simulated data, and run it on ICGC data. This is good work, and the manuscript is of interest to the readership of this journal. I have the following additional suggestions/comments.

1. Please state assumptions clearly. As stated above, I believe that there are two key assumptions, and that the other assumptions (inf sites, assumptions on CNA/SNV interplay, etc.) are secondary to these. It would be good if this is reflected in the writeup.

2. How robust is the method to violations of the assumptions? That is, please assess with simulations the following scenarios:

- * Sample is composed of clones from distinct branches.
- * Complex CNA/SNV scenarios: SNV was gained prior to CNA (e.g. amplification) and as such it may occur in varying numbers.
- * Loss of mutations/parallel evolution.
- * What happens when you change decrease/increase bin size?

3. Run on multi-region data. It'd be interesting to see what signature trajectories the method will infer for each region from the same patient. I believe that some patients in ICGC are multi-sample.

Minor

* Abstract should avoid using jargon that is defined only later in the manuscript (e.g. changepoint, signature exposure, activity).

(The term evolutionary trajectory was recently used in the following paper:

Caravagna, G., Giarratano, Y., Ramazzotti, D., Tomlinson, I., Graham, T. A., Sanguinetti, G., & Sottoriva, A. (2018). Detecting repeated cancer evolution from multiregion tumor sequencing data. *Nature Methods*, 15(9), 707–. <http://doi.org/10.1038/s41592-018-0108-x>

It'd be good to describe how/why TrackSig differs.

Reviewer #2:

Remarks to the Author:

This study developed a new tool, TrackSig, to reconstruct evolutionary trajectories of single tumors based on the signatures of mutational processes. Using the cancer cell fraction (CCF), the authors estimated an approximate order in which the somatic mutations accumulate, and inferred a trajectory of the mutational signature activities. The pipeline that the manuscript provided is user-

friendly, and easy to be tested. However, there are major concerns that remain to be addressed.

1) The high level of heterogeneity and the spatial structure within tumors have been sophisticatedly analyzed in literature, such as Gerlinger et al, Zhang et al, de Bruin et al, Ling et

al, Sottoriva et al, Yates et al, Harbst et al and so on. Apparently, the evolutionary trajectories of multiple clones within a tumor or among tumors in a patient can not be correctly traced by sequencing a bulk tumor sample, although the number of subclones and cancer cell fraction may be roughly estimated. If the phylogeny in tumors are reconstructed using the approaches that those previous studies have carried out, the activities of mutation signatures can be precisely timed and directly analyzed at the different stages (trunk and branches) during cancer evolution. In this case, there is no need to perform TrackSig. The novelty and improvement in understanding the evolutionary history of cancer cells are not convincingly significant in this study.

- [1] Gerlinger M, Rowan AJ, Horswell S, et al. Intratumor heterogeneity and branched evolution revealed by multiregion sequencing[J]. *New England Journal of Medicine*, 2012, 366: 883-892.
- [2] Zhang J, Fujimoto J, Zhang J, et al. Intratumor heterogeneity in localized lung adenocarcinomas delineated by multiregion sequencing[J]. *Science*, 2014, 346: 256-259.
- [3] de Bruin EC, McGranahan N, Mitter R, et al. Spatial and temporal diversity in genomic instability processes defines lung cancer evolution[J]. *Science*, 2014, 346: 251-256.
- [4] Gerlinger M, Horswell S, Larkin J, et al. Genomic architecture and evolution of clear cell renal cell carcinomas defined by multiregion sequencing[J]. *Nature Genetics*, 2014, 46: 225-233.
- [5] Ling S, Hu Z, Yang Z, et al. Extremely high genetic diversity in a single tumor points to prevalence of non-Darwinian cell evolution[J]. *Proceedings of the National Academy of Sciences of the United States of America*, 2015, 112: E6496-6505.
- [6] Sottoriva A, Kang H, Ma Z, et al. A Big Bang model of human colorectal tumor growth[J]. *Nature Genetics*, 2015, 47: 209-216.
- [7] Yates LR, Gerstung M, Knappskog S, et al. Subclonal diversification of primary breast cancer revealed by multiregion sequencing[J]. *Nature Medicine*, 2015, 21: 751-759.
- [8] Harbst K, Lauss M, Cirenajwis H, et al. Multiregion Whole-Exome Sequencing Uncovers the Genetic Evolution and Mutational Heterogeneity of Early-Stage Metastatic Melanoma[J]. *Cancer Research*, 2016, 76: 4765-4774.

2) Since TrackSig is designed to reconstruct evolutionary trajectories of mutations in cancer, the authors need to test its accuracy of estimating the order of the somatic mutations. In the literature that focused on tracing the evolutionary history within tumors, the order of mutation accumulation is quite clear. The published data in literature mentioned above could be used to do the test.

3) It is skeptical to infer the order of somatic mutations based on CCF. In the cases that the evolution of tumors has subject to positive selection, a younger subclone in which driver mutations have accumulated may obtain relatively high fitness, consequently, be bigger in size. Higher CCF of this subclone could be observed than the other subclones, resulting in a wrong order of mutation acceleration.

4) A test that the authors have ignored is to compare the performance of TrackSig to the published and widely used methods for inferring CCF and estimating the signature activity, such as ABSOLUTE (Carter et al., *Nat Biotechnol*, 2012), Pyclone (Roth et al., *Nat Methods*, 2014), Sclust (Cun et al., *Nat Protoc*, 2018), PhyloWGS (Deshwar et al., *Genome Biol*, 2015), and deconstructSigs (Rosenthal et al., 2016), which is critical for a study in technical development to state the improvement and innovation in technology or scientific interpretation in the related field.

5) To evaluate the performance of TrackSig, we carried out the comparison between the estimations of TrackSig and published software (see Performance test), using the associated data that the manuscript provided and an unpublished whole-exome sequencing data with high coverage of 1500X for a lung cancer (personal communication). However, it turns out that the results are not consistent, which is shown below (Test II).

Performance test

Test I

In the first test, the associated data that the manuscript provided were used. The performance is

consistent with the result presented in the manuscript.

Test II

Whole-exome sequencing data with sequencing depth of 1500X in a lung cancer sample were used to test the performance of TrackSig. The sequencing data are unpublished, upon the authorization of a collaborating project and collaborators. The input files included variant allele frequencies (VAFs) and copy number aberration (CNA) detected in this tumor sample. First, we estimated the order of SNVs according to cancer cell fraction (CCF). Next, we inferred a trajectory of the mutational signature activities over the estimated ordering of the SNVs.

TrackSig identified one change-point of signature activities, which was close to the boundary between the clonal and subclonal mutations. Signature 1, 2, 5, 6 and 17 were active in clonal mutations. The subclonal signatures were predominated by signature 5 and 6, while the activity of signature 1, 2 and 4 decreased in subclonal mutations (Figure 1).

Figure 1. Activity trajectory by TrackSig. (See attachment)

We re-analyzed the signature activities for the same sample to compare the performance between TrackSig and published software. We distinguished clonal and subclonal mutations using PyClone (Roth et al., 2014), and estimated signature activities using deconstructSigs (Rosenthal et al., 2016). The observed signature activity has few features in common with TrackSig. In clonal mutations, signature 5, 12, 13, 16, 19 and 30 were active, while the subclonal mutations were contributed by signature 1, 3, 8 and 9 (Figure 2).

Figure 2. Clonal and subclonal signatures revealed by PyClone and deconstructSigs (See attachment)

Reference:

- [1] Rosenthal R, McGranahan N, Herrero J, et al. DeconstructSigs: delineating mutational processes in single tumors distinguishes DNA repair deficiencies and patterns of carcinoma evolution[J]. *Genome Biology*, 2016, 17: 31.
- [2] Roth A, Khattra J, Yap D, et al. PyClone: statistical inference of clonal population structure in cancer[J]. *Nat Methods*, 2014, 11: 396-398.

Attached please find the entire comments, including the text and figures.

Reviewer #3:

Remarks to the Author:

In this manuscript a method to deconvolve mutational process activity (via mutational signatures) through tumour evolution is presented. The method, called TrackSig, works by estimating the cancer cell fraction (CCF) of each single nucleotide variant, ranking variants according to CCF, followed by binning of variants based on their rank, and then estimating mutational signature contributions in each bin. Finally, a statistical change-point methodology is applied to detect bins where signature activity changes significantly.

The primary application of TrackSig is described in this manuscript as being part of the tumour heterogeneity PACWG paper (ref 17), and this paper is described as only providing some supporting assessment of the statistical accuracy of TrackSig measurements. Consequently, it reads rather like a supplementary note (and perhaps should be a supplementary note for ref 17 in order to support the claims in that paper?).

The manuscript is a technical methods paper, with no new data analysis, and I am doubtful if it

would be of significant general interest. The method is incremental, as all of the constituent parts of the TrackSig are already published and/or are in widespread general use and so the novelty is only to bring the constituent parts together in a single unified software tool. Specifically, CCF calculations are routine (e.g. refs 1, 9, 10, 15, 19 and 20 cited by the authors), expectation-maximisation methods for signature estimation are well established (e.g. <https://doi.org/10.1186/gb-2013-14-4-r39>), there are numerous previous reports of signature analysis by VAF/CCF (many of the refs 1-20, though the analysis in those papers is cruder than the method presented in this manuscript), and change-point detection methods are long established for time-series data (some references from the authors provided). There is no new primary data in this manuscript, and the simulation tool used to evaluate the method is of questionable value (see below).

Comments:

1. From what is presented, it seems that the simulations only address the ability of the method to find change points in signature activity. In order to fully test the accuracy and robustness of their method, I strongly suggest that the authors also examine the very important 'real life' confounding effects of: (a) inaccuracy in CCF estimation (which is VAF dependent, e.g. CCF estimations are worse at low VAF), (b) sequence depth and cellularity, (c) inaccuracy of mutational process activity estimation in each CCF bin.

The description of the simulations is also rather lacking in detail, for instance: the number of simulations evaluated is not reported, nor how it was decided when mutational processes change activity.

2. The general idea of VAF/CCF as a clock for tumour evolution is somewhat controversial. A few recent papers suggest that CCF is a clock under neutral evolution, but it is widely accepted that clonal selection leads to characteristic 'clusters' in the CCF distribution representing distinct clones (e.g. refs 16 and 20 describe tools to find these clusters). The CCF of the clone is thought to be proportional to its selective advantage (e.g. highly selected clones are larger), invalidating the underlying premise of the TrackSig tool, that CCF~time (previous studies have sidestepped this issue by comparing signatures between clones, or simply clonal vs subclonal CCF). The goodness of the 'CCF clock' for temporal reconstruction of mutational process activity could be assessed by simulation.

3. Construction of a timeline. The timeline is constructed by ordering mutations according to their CCF, and then binning mutations into groups of 100 mutations. This method feels rather arbitrary, and more importantly, I think it risks distorting the true view of tumour evolution (in a manner related to the previous point). Specifically, if there are subclones in a tumour, then all the mutations that founded that subclone are expected to be at the same CCF, and thus splitting the CCF of these clusters and assuming slight differences in CCF represent temporal changes risks assigning biological significance to sequencing noise.

Relatedly, it is not clear to me why the method does not just partition of CCF directly (rather than partition on the ordered list). I suggest this alternative method is evaluated/contrasted in simulations.

How was the number $n=100$ mutations per bin chosen? Does $n=100$ give a suitably robust estimate of signature activity versus the tradeoff with temporal resolution? (a comment "data not shown" regarding this point is included in the manuscript).

4. Sequencing depth is also a limiting factor for mutational signature reconstruction based on CCF. Here, the authors suggest that depth is only an issue for CCF/VAF clustering based methods (intro, third-to-last paragraph), but I do not see why this is the case. The ability to resolve fine-grained temporal changes in signature activity in low-depth sequencing data (PCAWG is around 50X coverage I think) needs to be proven, and could be readily addressed by simulation. When sequencing data is presented (e.g. figure 1), I strongly suggest that the CCF distribution should be

shown adjacent to the inferred signature activities to give confidence in the accuracy of the CCF estimates.

5. Does Table 1 report the number of correctly found change-points, or just the proportion of simulations where the right number of change-points (that were possibly in the wrong place) were found? The former is more meaningful of course.

6. Throughout the manuscript only relative mutational process activities are reported. There is interest in understanding absolute activity (e.g. total number of mutations attributable to a mutational process), as the arrival of mutator phenotype can 'drown out' other signatures when only proportions are considered. It would be useful to include estimations (and assessment of accuracy) of absolute signature activity, perhaps as supplementary material.

We would like to thank the reviewers for their careful attention to our manuscript and their helpful feedback. Our manuscript has benefited considerably by responding to their comments. We have added an entirely new section to address concerns shared by multiple reviewers. This section includes extensive and comprehensive testing on simulated data of TrackSig and a baseline method: SciClone followed by DeconstructSigs. We have also made extensive edits to address other reviewer comments and concerns. To aid in review, we have included a version of our manuscript with new, or significantly altered, text highlighted in red.

Below we first broadly address these shared concerns, and then follow with a point-by-point response to each reviewer query.

General shared concerns

The three reviewers raised two key concerns that we have now addressed.

First, they wanted us to assess how sensitive TrackSig's reconstructions are to violations of our assumptions; although there was lack of clarity about what those assumptions are. We have now updated the text to clarify those assumption, and generated simulated VAF data to test TrackSig's reconstructions under various violations of clonal evolution theory. We are happy to report, in summary, that TrackSig's reconstruction are robust to strong violations of these assumptions; whereas the baseline method is not. Furthermore, TrackSig's signature activity reconstructions are much better than those of the baseline methods, and in some cases, it performs more accurate subclonal reconstruction. These simulations and our performance are described in a new section in our manuscript entitled **Clonal evolution simulations**.

Second, they wanted us to compare against a baseline reflecting previous approaches for detecting change in signature activities. As mentioned in the previous paragraph, we evaluated this baseline in the clonal evolution simulations. The baseline was similar to that suggested by Reviewer #2: we first cluster SNVs by their VAF into subclones using SciClone, and then we predict signature activities based on SNVs assigned to each detected subclone using DeconstructSigs. We used SciClone instead of PyClone for clustering because the latter was unable, in our hands, to cluster more than 1000 SNVs. We test this strategy in normal reconstructions, and in the violation scenarios referred to in the previous paragraph.

In summary, TrackSig's reconstructions of signature activity are considerably better than those of the baseline strategy, even when we provide the baseline strategy with the correct noise model for the VAFs. Also, although TrackSig is not designed to detect subclones, it does so much better than the baseline method.

We have made every effort to respond to other concerns and feedback below on a point-by-point basis.

Reviewer #1 (Remarks to the Author):

The authors introduce TrackSig, a method for inferring how mutation signatures have changed over time in individual tumors. The method takes as input (1) a set of SNVs with associated CCFs and (2) a set of active signatures, and outputs for each timepoint (a set of SNVs with similar CCFs) the abundance of each active signature. TrackSig is based on two key assumptions: (1) the tumor sample is composed of clones present from the same branch in the underlying phylogenetic tree, (2) CCF of an SNV is correlated with the time at which that SNV was introduced. The authors validate their method on simulated data, and run it on ICGC data. This is good work, and the manuscript is of interest to the readership of this journal.

We appreciate your compliment, thank-you!

I have the following additional suggestions/comments.

1. Please state assumptions clearly. As stated above, I believe that there are two key assumptions, and that the other assumptions (inf sites, assumptions on CNA/SNV interplay, etc.) are secondary to these. It would be good if this is reflected in the writeup.

We have tried to clarify our assumptions. It is tricky to do so, because TrackSig simply assumes that signature activity is a piecewise constant function of the SNV ordering in the timeline (ordering by their estimated CCF).

TrackSig is designed to reconstruct the activities and find change-points as the function of CCF. Strictly speaking, for correct reconstruction, we don't CCF to increase monotonically with the age of the SNV, we simply need SNVs with similar CCFs to be generated from similar signature activity profiles.

This latter assumption remains true when there is branched subclone; and now we have included results on simulated data that show that TrackSig performs well under these conditions.

We have now clarified this point in our manuscript. For ease of presentation, some sections still assume that CCF is correlated with time; rather than rewrite those sections, we now make it clear that this is not required for correct reconstruction of activity trajectories.

We've added a section to the manuscript clarifying this, which we reproduce below:

2.1 Ordering the SNVs

No single evolutionary model can yet explain all of the observed VAF distributions in bulk tumor samples^{21, 24, 25}; using a CCF-based ordering of SNVs allows TrackSig to track changes in signature activity under a variety of such models. The clonal evolution model of cancer²⁶ posits that SNVs belong to one of a handful of subclones whose associated mutations all have the same CCF. Under this model, mutations from different subclones would occupy distinct regions of the CCF space, so if signature activities differed between subclones, the segments detected by TrackSig will mark the presence of distinct subclones. In neutral evolution models (e.g.,^{21, 24}) that assume SNVs to be unique and persistent, i.e., the infinite sites assumption, SNVs with higher CCFs generally occur earlier in the tumor's evolution, so when SNVs are sorted in order of decreasing CCFs, TrackSig's trajectories track changes in signature activity over time. In the following sections, we describe how the SNV VAFs are used to create a timeline to which TrackSig is applied. For ease of presentation, we will assume that the time of SNV occurrence increases approximately monotonically with position on the timeline. This interpretation is valid under the infinite sites assumption and either a neutral evolution model or a clonal one when all subclones are from the same branch, as is often the case in single samples. In section 3.5.2, we test TrackSig's reconstruction accuracy on simulated data consistent with different evolutionary models and violations to these assumptions; and in section 4.2, we discuss how to interpret TrackSig's reconstructions when the timeline is not a faithful representation of time of SNV acquisition.

2. How robust is the method to violations of the assumptions? That is, please assess with simulations the following scenarios:

Excellent suggestion! We have done the recommended simulations and are happy to report that TrackSig is largely insensitive to reasonable violations of our assumptions. In contrast, a baseline strategy of running SciClone and then DeconstructSig, inspired by Reviewer #2, is very sensitive to model misspecification.

** Sample is composed of clones from distinct branches.*

See "branching" below in Figure 5(b)

** Complex CNA/SNV scenarios: SNV was gained prior to CNA (e.g. amplification) and as such it may occur in varying numbers.*

See "CNA" below

** Loss of mutations/parallel evolution.*

See "inf site viol" below

Figure caption (a) Median activity reconstruction error (absolute activity difference) between the method (TrackSig and SciClone) and the true activities on clonal evolution simulations. The results are shown for the simulations of depth 30. Mean activity error: TrackSig 0.035, SciClone 0.062. **(b)** Comparison on number of change-points detected by different methods. X-axis shows different simulation types. Y-axis show the percentage of simulations where the method predicted the correct number of change-points. The results are shown for the simulations of depth 30.}

In the above “SciClone” refers to the SciClone+DeconstructSigs pipeline using the default sampling noise model for SciClone.

* *What happens when you change decrease/increase bin size?*

TrackSig’s performance is largely insensitive to changes in bin size, within a reasonable range (25-500 mutations). Please see the included figures which are now Supplementary Figure 5.

Figure caption: Median absolute difference and KL divergence between true activities and activities estimated by TrackSig computed for different bin sizes. Note that the scale difference on Y axis is very small.

3. Run on multi-region data. It'd be interesting to see what signature trajectories the method will infer for each region from the same patient. I believe that some patients in ICGC are multi-sample.

We have added signature trajectories for all the PCAWG multi-sample cases in the supplement. The activities for the clonal clusters are consistent with one another, with only a 0.03 mean absolute activity difference and 0.017 KL divergence between signature activities of clonal cluster across different samples. As one would expect, there are some differences in trajectories among the subclones, including oscillations which would be consistent with branching subclones in a single sample.

Minor

** Abstract should avoid using jargon that is defined only later in the manuscript (e.g. changepoint, signature exposure, activity).*

We have updated the abstract to define these terms.

(The term evolutionary trajectory was recently used in the following paper:

Caravagna, G., Giarratano, Y., Ramazzotti, D., Tomlinson, I., Graham, T. A., Sanguinetti, G., & Sottoriva, A. (2018). Detecting repeated cancer evolution from multiregion tumor sequencing data. Nature Methods, 15(9), 707–. <http://doi.org/10.1038/s41592-018-0108-x>

It'd be good to describe how/why TrackSig differs.

Thank you for pointing out the recent work. Although the paper uses the same term “evolutionary trajectory”, our approach is rather different. TrackSig represents evolutionary trajectories of cancer through mutational signatures on a continuous and linear CCF scale. In contrast, *Caravagna et al, 2018* look at cancer evolution through phylogenetic trees.

This paper analyses data from multi-region sequencing; whereas TrackSig's reconstructions are based on data from a single sample which is more common. We do not cluster mutations by CCF; thus need not make as many assumptions about the evolutionary process that generated the VAF data. We have now added a section comparing TrackSig's approach to methods of this type and include this among those methods.

Reviewer #2 (Remarks to the Author):

Note: this Reviewer included a list of references to prior work in their review. All papers in that list are now referenced in the current manuscript and we thank the reviewer for pointing out this discrepancy. We have removed that list below but otherwise have included their entire review.

This study developed a new tool, TrackSig, to reconstruct evolutionary trajectories of single tumors based on the signatures of mutational processes. Using the cancer cell fraction (CCF), the authors estimated an approximate order in which the somatic mutations accumulate, and inferred a trajectory of the mutational signature activities. The pipeline that the manuscript provided is user-friendly, and easy to be tested. However, there are major concerns that remain to be addressed.

Thank-you for taking the time to download and run our software. We appreciate your compliment about its usability and are happy that it worked well for you!

1) *The high level of heterogeneity and the spatial structure within tumors have been sophisticatedly analyzed in literature, such as Gerlinger et al, Zhang et al, de Bruin et al, Ling et al, Sottoriva et al, Yates et al, Harbst et al and so on. Apparently, the evolutionary trajectories of multiple clones within a tumor or among tumors in a patient can not be correctly traced by sequencing a bulk tumor sample, although the number of subclones and cancer cell fraction may be roughly estimated.*

We agree that multiple, spatially disparate samples can, in some cases, provide more information about evolutionary trajectories. Our manuscript shows that tools like TrackSig can improve the accuracy of these evolutionary reconstructions from single bulk samples, which are much more common than those from multi-region sequencing. Our work, in addition to providing an extremely useful method, suggests that there is a lot more information available than anticipated by this earlier work which relied solely on allele frequencies.

If the phylogeny in tumors are reconstructed using the approaches that those previous studies have carried out, the activities of mutation signatures can be precisely timed and directly analyzed at the different stages (trunk and branches) during cancer evolution.

We believe that in many cases, the assignment of mutations to subclones (and trunk and branches) is unreliable and this interferes with signature activity reconstructions. We have now added simulations to our manuscript supporting this viewpoint: we performed mutational clustering using SciClone and then used DeconstructSigs to infer the signature activities. We show that, at least for a single sample, that this approach is less accurate than TrackSig.

In this case, there is no need to perform TrackSig. The novelty and improvement in understanding the evolutionary history of cancer cells are not convincingly significant in this study.

Please see above.

We agree that if it was possible to perform *perfect* phylogenetic reconstruction and grouping of mutations into subclones, it would make the approach used in TrackSig unnecessary. However, perfect reconstruction is impossible, multi-region WGS sequencing is rare, and our simulations have shown, convincingly, that TrackSig is a better predictor of signature activities when provided with only one WGS sample. Furthermore, the proposed strategy pre-assumes that the genetic diversity among the multiple samples is consistent with a clonal evolution theory, this claim is still controversial.

We note, for example, that most of these reconstructions have substantial uncertainty in their mutation assignments, which, as we now show, interferes with correct estimation of signature activities. Indeed, some (e.g. El-Kebir et al, Nature Genetics 2018) have claimed to find errors in published cancer phylogenies (e.g. Gudem et al, Nature 2015) by using additional data.

In contrast, TrackSig makes no assumptions about the existence of subclones (in other words, “clonal evolution theory”), and is robust to low sequencing depth which would make the identification of subclones nearly impossible.

For these reasons, compared to six state-of-the-art subclonal reconstruction methods used in PCAWG, TrackSig identifies 1,250 new subclonal boundaries not found in the consensus reconstruction (Dentro et al, bioRxiv 312041) . These reconstruction methods include those in the papers referred to below by the reviewer.

2) *Since TrackSig is designed to reconstruct evolutionary trajectories of mutations in cancer, the authors need to test its accuracy of estimating the order of the somatic mutations. In the literature that focused on tracing the evolutionary history within tumors, the order of mutation accumulation is quite clear. The published data in literature mentioned above could be used to do the test.*

As mentioned above, we have already performed the recommended comparison (Dentro et al, bioRxiv 312041). TrackSig is more sensitive to subclonal boundaries, indicating that it performs better at estimating the ordering. Also, we have performed extensive simulations with a baseline method that the reviewer recommended below. The results of this in silico analysis are consistent with those in the Dentro et al manuscript.

3) *It is skeptical to infer the order of somatic mutations based on CCF. In the cases that the evolution of tumors has subject to positive selection, a younger subclone in which driver mutations have accumulated may obtain relatively high fitness, consequently, be bigger in size. Higher CCF of this subclone could be observed than the other subclones, resulting in a wrong order of mutation acceleration.*

The reviewer is concerned about our analysis of samples where subclones from different branches of the phylogeny are represented.

As mentioned above, in our introduction and in our response to Reviewer #1, we have now tested TrackSig in recommended circumstance. Its activity reconstructions are accurate even when there are misordered branching subclones.

4) *A test that the authors have ignored is to compare the performance of TrackSig to the published and widely used methods for inferring CCF and estimating the signature activity, such as ABSOLUTE (Carter et al., Nat Biotechnol, 2012), Pyclone (Roth et al., Nat Methods, 2014), Sclust (Cun et al., Nat Protoc, 2018), PhyloWGS (Deshwar et al., Genome Biol, 2015), and deconstructSigs (Rosenthal et al., 2016), which is critical for a study in technical development to state the improvement and innovation in technology or scientific interpretation in the related field.*

The methods mentioned here perform different tasks; some do copy number reconstruction (ABSOLUTE, Sclust), others only cluster mutations based on VAF (Pyclone, Sclust), others do phylogenetic reconstruction (PhyloWGS). DeconstructSig is the only method mentioned that estimates signature activity like TrackSig, and it only does so for a predefined group of mutations. TrackSig is novel because it identifies changes in signature exposure which often mark boundaries between subclones. It does this without prior clustering of VAFs, so it is not directly comparable to these methods: none of them do what TrackSig does.

To address this concern more broadly, however, we have compared TrackSig to a pipeline recommended by the reviewer in which mutations are first grouped by VAF using SciClone and then signature activities are estimated using DeconstructSig.

5) *To evaluate the performance of TrackSig, we carried out the comparison between the estimations of TrackSig and published software (see Performance test), using the associated data that the manuscript provided and an unpublished whole-exome sequencing data with high coverage of 1500X for a lung cancer (personal communication). However, it turns out that the results are not consistent, which is shown below (Test II).*

Thank-you for performing these tests! We are happy to see that you were able to reproduce our results on a subset of the data we provide. Without having access to the data that you used for Test II, we are unable to diagnose why the unpublished pipeline that you used failed to reproduce the results that TrackSig generated.

To try to address your concern, as indicated in our response to Reviewer #1, we have built our own SciClone and DeconstructSig pipeline and applied it to simulated data consistent with the assumptions of SciClone. Note that we replaced Pyclone from your pipeline with SciClone because Pyclone, in our hands, was unable to cluster more than 1000 SNVs, whereas WGS data usually has many more SNVs than this.

The SciClone+DeconstructSig pipeline performs worse than TrackSig on average at estimating signature activity; and often performs worse at detecting subclones.

Performance test

Test I

In the first test, the associated data that the manuscript provided were used. The performance is consistent with the result presented in the manuscript.

Thank-you for letting us know and for reproducing our results. We are glad to see that you were successful at doing so.

Test II

Whole-exome sequencing data with sequencing depth of 1500X in a lung cancer sample were used to test the performance of TrackSig. The sequencing data are unpublished, upon the authorization of a collaborating project and collaborators. The input files included variant allele frequencies (VAFs) and copy number aberration (CNA) detected in this tumor sample. First, we estimated the order of SNVs according to cancer cell fraction (CCF). Next, we inferred a trajectory of the mutational signature activities over the estimated ordering of the SNVs.

TrackSig identified one change-point of signature activities, which was close to the boundary between the clonal and subclonal mutations.

We are also glad to see that this independent validation of TrackSig was successful.

Signature 1, 2, 5, 6 and 17 were active in clonal mutations. The subclonal signatures were predominated by signature 5 and 6, while the activity of signature 1, 2 and 4 decreased in subclonal mutations (Figure 1).

Figure 1. Activity trajectory by TrackSig. (See attachment)

We re-analyzed the signature activities for the same sample to compare the performance between TrackSig and published software. We distinguished clonal and subclonal mutations using Pyclone (Roth et al., 2014), and estimated signature activities using deconstructSigs (Rosenthal et al., 2016). The observed signature activity has few features in common with TrackSig. In clonal mutations, signature 5, 12, 13, 16, 19 and 30 were active, while the subclonal mutations were contributed by signature 1, 3, 8 and 9 (Figure 2).

Figure 2. Clonal and subclonal signatures revealed by Pyclone and deconstructSigs (See attachment)

The signatures that you reported to be found by TrackSig are more consistent with those reported to be active in lung in the COSMIC database than those that you report to be found by DeconstructSig. Please see the table for active signatures for Lung Adeno on COSMIC: <https://cancer.sanger.ac.uk/signatures/matrix.png>. As such, the differences between the activities here may be due to the failure of DeconstructSigs. As we now show in our manuscript, pipelines containing DeconstructSigs perform worse than TrackSig at reconstructing signature activities on simulated data.

Also, it appears that the reviewer has fit the full set of signatures to the Lung samples. We strongly suggest detecting a subset of active signatures before running TrackSig, or using cancer-specific signature (See Section 3.4 for details). One can use cancer-specific signatures from COSMIC by simply providing the cancer type of the sample, as described in Github instructions. Although estimating active signatures is not the part of TrackSig's contribution, we have an explanation for possible strategies in Section 3.4 and also provide code for estimating active signatures as a separate script.

Reference:

- [1] Rosenthal R, McGranahan N, Herrero J, et al. DeconstructSigs: delineating mutational processes in single tumors distinguishes DNA repair deficiencies and patterns of carcinoma evolution[J]. *Genome Biology*, 2016, 17: 31.
- [2] Roth A, Khattra J, Yap D, et al. PyClone: statistical inference of clonal population structure in cancer[J]. *Nat Methods*, 2014, 11: 396-398.

Attached please find the entire comments, including the text and figures.

Reviewer #3 (Remarks to the Author):

In this manuscript a method to deconvolve mutational process activity (via mutational signatures) through tumour evolution is presented. The method, called TrackSig, works by estimating the cancer cell fraction (CCF) of each single nucleotide variant, ranking variants according to CCF, followed by binning of variants based on their rank, and then estimating mutational signature contributions in each bin. Finally, a statistical change-point methodology is applied to detect bins where signature activity changes significantly.

The primary application of TrackSig is described in this manuscript as being part of the tumour heterogeneity PACWG paper (ref 17), and this paper is described as only providing some supporting assessment of the statistical accuracy of TrackSig measurements. Consequently, it reads rather like a supplementary note (and perhaps should be a supplementary note for ref 17 in order to support the claims in that paper?).

Significantly methodological contributions, such as TrackSig, are often published separately from main papers where they are used. In fact, the heterogeneity PCAWG paper has already given rise to other manuscripts based on methods developed for it, e.g. Sclust (PMID: 29844525).

The manuscript is a technical methods paper, with no new data analysis, and I am doubtful if it would be of significant general interest.

We note that Reviewer #1 disagrees with this view, as does the steering committee of PCAWG -- who all reviewed this manuscript already. And furthermore, despite being unpublished, our GitHub repository already has 22 stars (none from Morrislab) and six forks from outside Morrislab, and our bioRxiv manuscript has been cited twice, once in a review (PMID: 30041675) and once in a prostate cancer genomic paper in Cell. Once published in a prominent journal like Nature Communications, we expect continuing broad interest in our work.

The method is incremental, as all of the constituent parts of the TrackSig are already published and/or are in widespread general use and so the novelty is only to bring the constituent parts together in a single unified software tool.

We thank the reviewer for pointing out the novelty of our method. We agree that there is no method or software to perform the analysis that TrackSig does. As a result, we are the first to discover that subclones can be identified simply on the basis of this change in mutational signatures. We further note, that we have now added extensive simulations showing that previously described approaches to attempting to detect change in signature activities (i.e. the SciClone + deconstructSig pipeline) perform badly. This may be why our observation was previously unreported.

Specifically, CCF calculations are routine (e.g. refs 1, 9, 10, 15, 19 and 20 cited by the authors), expectation-maximisation methods for signature estimation are well established (e.g. <https://doi.org/10.1186/gb-2013-14-4-r39>),

We have added a citation to this relevant work. Thank-you for directing us to it, and we are embarrassed that we had overlooked it.

there are numerous previous reports of signature analysis by VAF/CCF (many of the refs 1-20, though the analysis in those papers is cruder than the method presented in this manuscript),

Thank-you for pointing out the increased precision of our approach. All of this prior work clusters mutations first by VAF before doing the signature analysis. As we have shown, this crude strategy causes subclones to be missed (as shown in our simulations, and in Dentre et al) and signature activities to be badly estimated. Furthermore, clustering by VAF implicitly assumes clonal evolution theory; however, this assumption is inappropriate for up to 30% of tumour

samples (Williams et al, Nature Genetics 2016), i.e. when the VAF data are consistent with a neutrally evolving tumour.

and change-point detection methods are long established for time-series data (some references from the authors provided).

We have not seen the particular changepoint detection method we employ being used in the cancer field before. Indeed, commonly used methods, like binary segmentation, are shown in the PELT manuscript to be highly inaccurate.

However, more generally, we would like to point out that all new algorithms are built on the foundation of previous work. Like all new statistical algorithms, we have put together prior work in a creative and novel way. In this case, we are the first to use change-point finding on mutation types for (i) signature analysis; and (ii) for finding subclones. None of the prior work referred to by the reviewers or in our paper, made the crucial observation that this would be possible. So, we do not understand the reviewers concern here.

There is no new primary data in this manuscript, and the simulation tool used to evaluate the method is of questionable value (see below).

We have added new simulations to address this concern. Please see our introduction and responses above.

Comments:

1. *From what is presented, it seems that the simulations only address the ability of the method to find change points in signature activity. In order to fully test the accuracy and robustness of their method, I strongly suggest that the authors also examine the very important 'real life' confounding effects of: (a) inaccuracy in CCF estimation (which is VAF dependent, e.g. CCF estimations are worse at low VAF), (b) sequence depth and cellularity, (c) inaccuracy of mutational process activity estimation in each CCF bin.*

Please see our responses to Reviewer #1 and the new section on clonal evolution simulations designed to address these concerns.

The description of the simulations is also rather lacking in detail, for instance: the number of simulations evaluated is not reported, nor how it was decided when mutational processes change activity.

We have now added these details to the manuscript and we have a substantive supplement describing our new simulations in detail.

2. *The general idea of VAF/CCF as a clock for tumour evolution is somewhat controversial. A few recent papers suggest that CCF is a clock under neutral evolution, but it is widely*

accepted that clonal selection leads to characteristic 'clusters' in the CCF distribution representing distinct clones (e.g. refs 16 and 20 describe tools to find these clusters). The CCF of the clone is thought to be proportional to its selective advantage (e.g. highly selected clones are larger), invalidating the underlying premise of the TrackSig tool, that CCF~time (previous studies have sidestepped this issue by comparing signatures between clones, or simply clonal vs subclonal CCF). The goodness of the 'CCF clock' for temporal reconstruction of mutational process activity could be assessed by simulation.

This concern arises out of a misunderstanding of the assumptions underlying TrackSig. We have now clarified those assumptions, and in particular, directly address this concern regarding the accuracy of the CCF clock assumption, which is shared with Reviewer #1. Please see the text we added there.

3. *Construction of a timeline. The timeline is constructed by ordering mutations according to their CCF, and then binning mutations into groups of 100 mutations. This method feels rather arbitrary,*

We have now tested the robustness of TrackSig's reconstructions to changing the size of the bins. Its accuracy is largely insensitive to bin size.

and more importantly, I think it risks distorting the true view of tumour evolution (in a manner related to the previous point). Specifically, if there are subclones in a tumour, then all the mutations that founded that subclone are expected to be at the same CCF, and thus splitting the CCF of these clusters and assuming slight differences in CCF represent temporal changes risks assigning biological significance to sequencing noise.

Note that when constructing the timeline, we sample CCF from Beta distribution implied by the VAF to avoid the quantization artifacts that the reviewer is concerned with.

Indeed, in the ideal conditions at high coverage the CCF of mutations of the same subclones are assumed to be the same, assuming that the tumour evolved in manner consistent with clonal evolution theory. If CCF could be measured with *perfect* accuracy, there wouldn't be a need for methods for subclonal reconstruction. However, in real data, even at 50x coverage, CCFs of mutations from different subclones overlap in CCF space due to finite sample estimates of VAF. These factors make detecting subclones a difficult task, and they make assigning mutations accurately to subclones impossible.

If mutations from the same subclone are assigned to different bins, we would expect TrackSig to group the bins together because they would have a similar distribution of mutation types. Indeed, in support of this, we see that TrackSig's reconstructions are relatively insensitive to bin size (see below).

Relatedly, it is not clear to me why the method does not just partition of CCF directly (rather than partition on the ordered list). I suggest this alternative method is evaluated/contrasted in simulations.

We are uncertain whether the reviewer is asking us to (a) cluster mutations by CCF first, before reconstructing signature activities, or if (b) they are suggesting that we shrink the bin size to a single mutation.

If it is (a), this concern is addressed by our clonal evolution simulations. The reason we don't cluster by CCF first, is that this approach (i.e. clustering using SciClone) is less sensitive to changes particularly when the assumptions of clonal evolution theory are violated, and leads to inaccuracies in the estimation of signature activity, possibly because it performs hard assignment of mutations to subclones.

As we are sure the reviewer is aware, clustering by CCF requires making an assumption about the sequencing noise -- there is still some controversy about what the correct noise model is. Furthermore, clustering by CCF also assumes the claim that you made above, i.e., that the observed CCF distribution reflects adding sequencing noise to a handful of latent CCF values, one for each subclone. This is called clonal evolution theory. TrackSig need not make assumptions about either sequencing noise distributions, nor about clonal vs neutral evolution when doing its reconstructions.

If it is (b), then the reason we do not use a bin size of one because, as we now show, the sensitivity and accuracy of our methods is not very sensitive to changes in bin size down to 25. Using a bin size of 100 permits TrackSig to run more quickly because our time complexity depends on the number of bins -- so 100x fewer bins makes TrackSig run at least 100x faster.

How was the number $n=100$ mutations per bin chosen? Does $n=100$ give a suitably robust estimate of signature activity versus the tradeoff with temporal resolution? (a comment "data not shown" regarding this point is included in the manuscript).

Yes. We have tested TrackSig with bin sizes 25, 50, 75, 100, 150, 200, 300, 500. As the figure below shows, smaller bins lead to similar accuracies in activity reconstructions.

Figure caption: Median absolute difference and KL divergence between true activities and activities estimated by TrackSig computed for different bin sizes. Note that the scale difference on Y axis is very small.

4. *Sequencing depth is also a limiting factor for mutational signature reconstruction based on CCF. Here, the authors suggest that depth is only an issue for CCF/VAF clustering based methods (intro, third-to-last paragraph), but I do not see why this is the case. The ability to resolve fine-grained temporal changes in signature activity in low-depth sequencing data (PCAWG is around 50X coverage I think) needs to be proven, and could be readily addressed by simulation. When sequencing data is presented (e.g. figure 1), I strongly suggest that the CCF distribution should be shown adjacent to the inferred signature activities to give confidence in the accuracy of the CCF estimates.*

We have now added clonal evolution simulations at a depth of 30x, which is standard in the PCAWG dataset, and 10x and 100x to address this concern (Figures F.2 and F.3 in the supplement). With a few exceptions, TrackSig is largely insensitive to changes in read depth in the accuracy of its reconstructions.

5. *Does Table 1 report the number of correctly found change-points, or just the proportion of simulations where the right number of change-points (that were possibly in the wrong place) were found? The former is more meaningful of course.*

Table 1 reports the number of correctly placed change-points.

This is clarified in section 3.5.1 “Non-parametric simulations”:

“To compute number of false positives and false negatives, we count a true positive detection if at least one of predicted change-points occur with three time points of an actual one. A false negative is when no predicted change-points are within three time points of an actual change. This criteria is identical to the one we use to evaluate whether a change-point supports a subclonal boundary. We deem a predicted change-point a false positive if it occurs more than three time points away from the closest actual change-point.”

6. *Throughout the manuscript only relative mutational process activities are reported. There is interest in understanding absolute activity (e.g. total number of mutations attributable to a mutational process), as the arrival of mutator phenotype can 'drown out' other signatures when only proportions are considered. It would be useful to include estimations (and assessment of accuracy) of absolute signature activity, perhaps as supplementary material.*

Each time point contains 100 mutations, so to get absolute estimates of the number of SNVs from each process in each time point, one simply needs to multiply TrackSig's activity levels by 100. We don't see the value in putting this simple calculation in the supplement.

We suspect, however, that the reviewer is asking us to estimate mutation rate / time for each process, under the assumption that this rate is constant for some processes but increases for others in a mutator phenotype. Doing so, requires us to be able to assign an absolute time estimate to the timeline; it is not yet clear how to do that. One possibility might be to use some sort of molecular clock -- e.g. the number of CG->TG mutations -- however, the accuracy and reliability of this clock is still controversial. We think that this is a very interesting question to be addressed in future work; in fact, we have spent some time on it early in the project but found it difficult to come up with an reliable estimator.

Reviewers' Comments:

Reviewer #1:

Remarks to the Author:

The authors have satisfactorily addressed my comments. This paper will be of interest to the readership of this journal, presenting some novel ideas in the clonal dynamics of mutational signatures. There is interest in the community about this (e.g. workshop on mutational signatures at the upcoming Pacific Symposium of Bioinformatics.)

Mohammed El-Kebir

Reviewer #3:

Remarks to the Author:

Some concerns remain after the response of the authors to my previous critique.

1. (related to previous point 1) The new inclusion of simulations of clonal evolution is welcome. However, the new simulation tool does not seem to adequately model the complexities of real data. Specifically:

a) it seems all clones are modelled as 'isogenic' (mutations all at the same CCF). This neglects within-clone (neutral) mutations, which all accrue at low frequency, and are expected to impact the inference of signature activity. I suggest a branching process is used to simulate tumour evolution (a number of branching process simulators have been published).

b) I couldn't see an explicit assessment of signature activity inference accuracy by sequencing depth. This is essential to include, as it would demonstrate the limits of inference possible in 35X coverage data, and potentially motivate experiments for higher depth sequencing. Figures F2 and F3 seem to suggest that near-perfect recovery of subclonal mutational signatures is possible even in 10X sequencing data. I find it hard to believe that this is correct, and suspect some inaccuracy in the simulations. 10X coverage is five mutant reads for clonal mutations in diploid cells, and so given the dispersion generated by binomial sampling it would not be possible to determine if any individual mutation is subclonal or not, let alone determine a mutation's CCF!

2. (related to previous point 3). Simulations testing the effect of bin size are now included, and seem to show bin size is unimportant. However, the data shows a strange 'spike' in divergence (rebuttal figure) at bin size ~50. Why is this? It seems rather odd. Presumably bin size and sequencing depth have interrelated effects (because lower depth sequencing means less accurate CCF) and so preferably these two features should be examined together.

3. (Related to previous point 3) There was some confusion about my previous comment about CCF partitioning - apologies for the lack of clarity. My concern remains, and derives from the statement "TrackSig first partitions the ordered mutations into bins of 100 mutations and interprets each bin as one time point. The timeline of the cancer is the collection of the time points." It is not clear to me why bins are constructed by counting sets of 100 mutations, rather than by predefining a series of fixed intervals on the CCF line (i.e. CCF 0-0.1, 0.1-0.2, 0.2-0.3, etc) and considering the mutations in each CCF bin. The latter idea feels more intuitive to this reviewer in relation to the idea that CCF is often proportional to time, and somewhat mitigates the concern about splitting mutational clusters based on (noisy) CCF inferences. (I note that the authors now perform CCF clustering as an alternative method too, which is welcome)

4. (related to previous point 4) The authors have ignored my request to show CCF distributions alongside mutational signature activity. I strongly urge them to do this so that the quality of CCF inference, and signature activity matching can be easily judged by the reader. I think CCF distributions are necessary in Figure 1, and hope they can be included for some example clonal evolution simulations in Figure 4.

5. (Related to previous point 6). Absolute signature activity. I see the confusion here - absolute signature activity only has meaning in the CCF bin method I suggest in the previous point, and I agree with the authors that it isn't interesting in the strict "100 mutations per bin" method. If the authors do investigate CCF binning as per my previous point (I hope that they do) then absolute signature activity (and accuracy of minority signature inference in the bin) would be worth investigating.

6. The analysis of PAWCG data is very cursory. For example, Figure 6 could be analysed by individual mutational process, by tumour type, by mutational burden etc etc.

We would like to thank the two remaining reviewers for their helpful comments. Reviewer #3 (R3)'s feedback has improved our manuscript and we thank them for their careful and thoughtful attention.

Reviewer #1 was satisfied with our response and also judged that we had fully addressed Reviewer #2's concerns.

We have addressed R3's concerns as follows:

1. We have added new histograms showing the mutation counts per signature in fixed width intervals of CCF. (Fig 1, 4)
2. We have added new simulations that incorporate neutral mutations, and evaluated TrackSig and the baseline method on them. (Section C.5, Figs 4, F.2, F.8)
3. We have modified our clonal evolution simulations, so that mutations with less than three variant reads are removed, this is in response to some concerns raised by R3. These changes resulted in some performance changes but TrackSig still outperforms the baseline method.
4. We have updated some figures (Figs 1, F.5, F.6) and added some new ones to address other concerns by R3 (Figs 4, F.7, F.8)

Below we respond to R3's comments point-by-point.

Reviewer #3 (Remarks to the Author):

Some concerns remain after the response of the authors to my previous critique.

Apologies. As we indicate below, it was because we misunderstood them. Thank-you for clarifying.

1. (related to previous point 1) The new inclusion of simulations of clonal evolution is welcome. However, the new simulation tool does not seem to adequately model the complexities of real data. Specifically:

a) it seems all clones are modelled as 'isogenic' (mutations all at the same CCF). This neglects within-clone (neutral) mutations, which all accrue at low frequency, and are expected to impact the inference of signature activity. I suggest a branching process is used to simulate tumour evolution (a number of branching process simulators have been published).

This is a good point. We have now added new simulations that include neutral mutations in each clone. To do so, we computed the number of mutations per subclone using the model of (Williams et al, 2016); and sampled their VAFs using their power law distribution.

The results are shown in Figs 3, F.2, F.8. These neutral mutations only have a substantial impact on TrackSig's reconstruction accuracy at 100x when there is a subclonal population. In this circumstance, the neutral mutations comprise more than 10% of the detected mutations and

form a new “neutral mode” near the VAF detection limit whose mutation type composition matches the clonal population, and not the subclonal population. When there are no subclones, TrackSig correctly ignores this “neutral mode” because its mutation type composition matches that of the clonal population.

Note that we used a direct approach to sample neutral mutations rather than a branching process as recommended by the reviewer. Our one cluster case precisely matches the neutral model described in Williams et al 2016; as such, we use the model that they derived to set our neutral mutation count and to sample their CCFs.

For the two cluster simulations, we used the same neutral model. Here it is less appropriate but it is still suitable for our needs because it establishes a lower bound on performance. Namely, in the two cluster case, if we were able to generate data with a subclone and neutral mutations from both lineages, the number of neutral mutations would be smaller than the number we used, though their VAF distributions would not drastically differ, and the “neutral mode” near the detection limit would be composed of mutations from both clones rather than from just one clone. These differences would make the reconstruction problem easier for TrackSig. As such, although the neutral model is not perfect in the two cluster case, the one we used provides lower bound on TrackSig’s performance.

We took this direct approach so that the new simulations would be comparable to the isogenic ones. This is important because, besides Williams et al (NG 2018) and Sun et al (NG 2017), all other subclonal reconstruction tools have been evaluated using the isogenic conditions. Thus, one can view our work as helping to illustrate the performance loss of the parametric clustering methods when run on simulations that include neutral mutations.

b) I couldn't see an explicit assessment of signature activity inference accuracy by sequencing depth. This is essential to include, as it would demonstrate the limits of inference possible in 35X coverage data, and potentially motivate experiments for higher depth sequencing.

Please see newly added figures F.7, F.8 . We had previously reported these accuracies (for 10x, 30x, and 100x) in different places in the manuscript (e.g., in the caption of Figure F.4) for the isogenic simulations. Figure F.7 now summarizes these in a single place; and we have added F.8 to compare reconstruction accuracies with neutral mutations.

Figures F2 and F3 seem to suggest that near-perfect recovery of subclonal mutational signatures is possible even in 10X sequencing data. I find it hard to believe that this is correct, and suspect some inaccuracy in the simulations. 10X coverage is five mutant reads for clonal mutations in diploid cells, and so given the dispersion generated by binomial sampling it would not be possible to determine if any individual mutation is subclonal or not, let alone determine a mutation's CCF!

Thanks for pointing this out, it is something we had overlooked and we now have changed our simulations and evaluations so that they are more realistic. The changes we made to make our performance estimates at 10x more accurate are i) we now remove mutations below the VAF detection threshold (which we set to be three variant reads) and ii) we no longer choose the best of two different parametric noise models for our SciClone baseline on a case-by-case basis.

First, it is important to point out that detecting whether a subclone exists doesn't require us to be able to confidently assign any mutations to that subclone. For example, in TrackSig, if there are detectable differences in the type distributions for mutations, for example, with three variant reads versus those with five, that would be enough to detect a changepoint. And we do see that in ~60% of the two cluster clonal evolution simulations at 10x (Figure F.2), we can recover the changepoint. We have added Figure 4b showing one such case, as requested. Even though there is only a single mode, there are differences in the mutation type distributions at the left and right tails, which are sufficient for TrackSig to detect a changepoint.

Parametric clustering models, like our SciClone baseline, can also correctly recover a second cluster even when the modes are overlapping, but that depends on having the correct noise model. As such, their reconstruction accuracies are extremely sensitive to model mis-specification error as we now show in Fig 3b and Fig F.2. In our previous manuscript, we had selected the best performing of the two noise models, which artificially inflates their performance. We now show accuracies for both models.

2. (related to previous point 3). Simulations testing the affect [sic] of bin size are now included, and seem to show bin size is unimportant. However, the data shows a strange 'spike' in divergence (rebuttal figure) at bin size ~50. Why is this? It seems rather odd.

We have replaced this plot with box plots (F.5 and F.6) showing the distribution of errors across the various examples, that apparent spike was a negligible difference and it disappeared when we slightly modified the simulations by removing mutations below the VAF threshold. We now use a y-axis range which includes 0 to better illustrate the lack of meaningful differences in performance between different bin sizes.

Presumably bin size and sequencing depth have interrelated effects (because lower depth sequencing means less accurate CCF) and so preferably these two features should be examined together.

Our original simulations were done at 100x (F.6), this is where we expected the greatest influence of bin size because here the CCF estimates are the most accurate. Even at this depth, we do not see any substantial influence of bin size on reconstruction accuracy. At lower read depth, the CCF estimates are less accurate, so the observed signature activities (as a function of CCF) change more smoothly and bin size would have even less of an impact. To illustrate this, we have now also added bin size simulations at 30x (F.5).

3. (Related to previous point 3) There was some confusion about my previous comment about CCF partitioning - apologies for the lack of clarity.

Thank-you for this clarification.

My concern remains, and derives from the statement "TrackSig first partitions the ordered mutations into bins of 100 mutations and interprets each bin as one time point. The timeline of the cancer is the collection of the time points." It is not clear to me why bins are constructed by counting sets of 100 mutations, rather than by predefining a series of fixed intervals on the CCF line (i.e. CCF 0-0.1, 0.1-0.2, 0.2-0.3, etc) and considering the mutations in each CCF bin. The latter idea feels more intuitive to this reviewer in relation to the idea that CCF is often proportional to time, and somewhat mitigates the concern about splitting mutational clusters based on (noisy) CCF inferences. (I note that the authors now perform CCF clustering as an alternative method too, which is welcome)

Stable estimation of signature activities requires a minimum number of mutations. As such, we can't use a fixed CCF interval because many intervals would contain too few mutations to allow us to estimate signature activities. Also, fixed width intervals limit the resolution of our CCF boundaries. Currently, the resolution is determined by the model's ability to estimate signature activities, so if the CCF estimates are very accurate, we can resolve small differences in CCF between subclones.

We have found a different way of addressing the reviewer's concerns regarding cluster splitting. We now allow users to set the bin size as low as 1; but we constrain segments to have at least 100 mutations -- i.e., if each time point contains only one mutation, each segment must contain at least 100 time points. This avoids cluster splitting, because segment boundaries could then occur between any two mutations. But it does limit the minimal number of mutations per segment to 100.

We were already constraining segments to contain at least four time points; all we have done is to make this parameter depend on the bin size. We will also make it accessible to users, if they want to ensure larger segments.

Note that neither the difficulty of estimating subclonal CCFs with high resolution, nor of having clusters with a small number of mutations, is unique to TrackSig. In a parametric model of mutation VAF, the stderr of subclonal CCF estimates is inversely proportional to \sqrt{n} , where n is the number of mutations assigned to a subclone. When n is low, there is a lot of uncertainty in the subclone's CCF. This, coupled with the high possibility of model mis-specification (i.e. due to incorrectly modeled observation noise distributions), means that it is very difficult to estimate the location of CCF boundaries in mutation poor regions with these models. We see this in the weak performance of the baseline models in our simulations.

4. (related to previous point 4) The authors have ignored my request to show CCF distributions alongside mutational signature activity.

Apologies, we actually do show this but it might not have been clear that we were doing so. The vertical lines in Figure 1, 4, and in the supplemental figures, show the values of the mean CCFs in each bin, as explained in the legend of Figure 1 in the current and previous versions of the manuscript. The density of these vertical lines indicates the distribution of the CCFs.

I strongly urge them to do this so that the quality of CCF inference, and signature activity matching can be easily judged by the reader. I think CCF distributions are necessary in Figure 1, and hope they can be included for some example clonal evolution simulations in Figure 4.

Now that we understand what the reviewer was asking, we are able to make the requested figure. It is a useful visualization, thanks for the suggestion! See Figures 1 and 4. Please note that we are not attempting to infer CCFs of individual mutations but rather the distribution of mutation types as a function of CCF. Because of this, when assigning CCF to a mutation we actually add Beta noise to its measured VAF to represent our uncertainty in the actual, hidden VAF. If we were trying to accurately estimate CCF for individual mutations, we would not be adding this noise.

5. (Related to previous point 6). Absolute signature activity. I see the confusion here - absolute signature activity only has meaning in the CCF bin method I suggest in the previous point, and I agree with the authors that it isn't interesting in the strict "100 mutations per bin" method. If the authors do investigate CCF binning as per my previous point (I hope that they do) then absolute signature activity (and accuracy of minority signature inference in the bin) would be worth investigating.

Please see the new versions of the Figures 1 and 4 described above.

6. The analysis of PAWCG data is very cursory. For example, Figure 6 could be analysed by individual mutational process, by tumour type, by mutational burden etc etc.

Interesting suggestions! However, they are beyond the scope of this work. Indeed, some of these suggestions would start to overlap work of the entire working group which is presented elsewhere. We also note that this is a new concern which did not appear in the initial reviews, nor does it appear in response to any of the changes that we have made.

Reviewers' Comments:

Reviewer #3:

Remarks to the Author:

I am broadly satisfied with the responses and modifications to the manuscript.

That tracksig is apparently effective in very low depth data is interesting and perhaps unexpected, and in general shows the power of considering an ensemble of mutations for evolutionary inference. I'm glad the authors investigated this.

One issue that still remains: The inclusion of neutral mutations using the analytic form $(1/f^2)$ from Williams NG 2016 is welcome, but the approach that the authors have taken is suboptimal, as they say "After discussion with our colleagues from population genetics, we decided to generate neutral mutations only from a clonal cluster." Neutral mutations are always generated within clones, so allowing neutral mutations only from the clonal cluster is simply incorrect. It is important for this manuscript, as within-clone neutral mutations confound the CCF~time assumption.

My preference remains that this addressed using a branching process simulator. If this is judged beyond the scope of the manuscript, I suggest adding a caveat along the lines of "the true complexity of tumour evolution [...details...] may confound the analysis in a way that is not assessed here, and so care must be taken interpreting results in tumours with complex clonal architectures".

Dear all,

Thank-you for the quick response. Our remaining reviewer was satisfied with our revision except for one remaining concern which we have addressed by selecting one of the two choices that the reviewer gave us, and adding precisely the text that they requested to our manuscript.

I am broadly satisfied with the responses and modifications to the manuscript.

Thank-you for your review.

That tracksig is apparently effective in very low depth data is interesting and perhaps unexpected, and in general shows the power of considering an ensemble of mutations for evolutionary inference. I'm glad the authors investigated this.

One issue that still remains: The inclusion of neutral mutations using the analytic form ($1/f^2$) from Williams NG 2016 is welcome, but the approach that the authors have taken is suboptimal, as they say "After discussion with our colleagues from population genetics, we decided to generate neutral mutations only from a clonal cluster." Neutral mutations are always generated within clones, so allowing neutral mutations only from the clonal cluster is simply incorrect. It is important for this manuscript, as within-clone neutral mutations confound the CCF-time assumption.

We have removed the quoted statement from the supplement and replaced it with text below. This text explains why we chose not to sample neutral mutations from the subclonal cluster in our simulations. As this new text makes clear, we are aware that our simulations are incorrect but we argue that they establish a lower bound on performance. This lower bound is all we require for this manuscript.

Note that we have already addressed questions about the validity of the CCF-time assumption in prior reviews and in prior versions of the manuscript. We are uncertain why this issue is now reappearing in review as this was addressed in our first response.

New text:

Note that the approach we used to sample neutrally-evolving mutations may not reflect the true, complex clonal dynamics that would be better represented with a branching process. Although our one cluster case precisely matches a standard neutral model, using the same model for the two cluster simulations ignores the effect that the introduction of subclone has on the number and VAF distribution of neutrally-evolving mutations.

It does, however, establish a lower bound on performance. The introduction of a subclone is likely to reduce the number of neutral mutations, though their VAF distributions would not drastically differ, and the "neutral mode" near the detection limit would be composed of mutations from both clones rather than from just one clone. These differences would make the

reconstruction problem easier for TrackSig. As such, although the neutral model is not correct in the two cluster case, the one we used provides lower bound on TrackSig's performance.

My preference remains that this addressed using a branching process simulator. If this is judged beyond the scope of the manuscript, I suggest adding a caveat along the lines of “the true complexity of tumour evolution [...details...] may confound the analysis in a way that is not assessed here, and so care must be taken interpreting results in tumours with complex clonal architectures”.

We have opted for the latter and have added the following text to our interpretation of the results:

Note, however, that this simulation may not be representative of real data because, unlike other simulations, we are not simulating neutral mutations from the subclone.

And added precisely the text suggested by the reviewer to our discussion, specifically:

Note, however, that the true complexity of tumour evolution with multiple subclones and high depth sequencing may confound the analysis in a way that is not assessed here, and so care must be taken interpreting results in tumours with complex clonal architectures.

We note that there is no established standard for which branching processes to use, nor what the correct parameterizations of that process are. As such, for these reasons and for the reasons we outlined in the previous revision (and now include for our readers in the supplement), a branching process simulation would be beyond the scope of this manuscript.